# Seeing but Not Believing: Probing the Disconnect Between Visual Attention and Answer Correctness in VLMs

**Zhining Liu[1], Ziyi Chen[1], Hui Liu[2], Chen Luo[2], Xianfeng Tang[2], Suhang Wang[2], Joy Zeng[2], Zhenwei Dai[2], Zhan Shi[2], Tianxin Wei[1], Hanqing Lu[2], Benoit Dumoulin[2], Hanghang Tong[1]**
[1]University of Illinois Urbana-Champaign, [2]Amazon
liu326@illinois.edu

## Abstract

Vision-Language Models (VLMs) achieve strong results on multimodal tasks such as visual question answering, yet they can still fail even when the correct visual evidence is present. In this work, we systematically investigate whether these failures arise from not *perceiving* the evidence or from not *leveraging* it effectively. By examining layer-wise attention dynamics, we find that shallow layers focus primarily on text, while deeper layers sparsely but reliably attend to localized evidence regions. Surprisingly, VLMs often perceive the visual evidence when outputting incorrect answers, a phenomenon we term *"seeing but not believing"* that widely exists in major VLM families. Building on this, we introduce an inference-time intervention that highlights deep-layer evidence regions through selective attention-based masking. It requires no training and consistently improves accuracy across multiple families, including LLaVA, Qwen, Gemma, and InternVL. These results show that VLMs encode reliable evidence internally but underutilize it, making such signals explicit can bridge the gap between perception and reasoning, advancing the diagnostic understanding and reliability of VLMs.

## 1 Introduction

Vision-Language Models (VLMs) (GeminiTeam et al., 2023; Achiam et al., 2023; Grattafiori et al., 2024; Bai et al., 2025) have recently achieved remarkable progress across a wide spectrum of multimodal tasks that require reasoning over both images and text. Among these tasks, Visual Question Answering (VQA) has been a central task for evaluating VLMs' ability to integrate and reason over visual and linguistic information (Kim et al., 2025; Zhang et al., 2024).

Despite these advances, recent studies have highlighted a persistent and puzzling gap between the availability of visual evidence in an image and the correctness of VLM answers (Wang et al., 2024; Kamoi et al., 2024). Specifically, models often overlook, ignore, or underutilize the crucial visual information, leading to errors even when the correct evidence is present in the image. This raises a fundamental question: Do VLMs fail in those cases because they cannot *perceive* the visual information, or because they fail to effectively *leverage* it in reasoning and generation?

Prior works have begun to probe this disconnect. Tong et al. (2024) showed that multimodal LLMs/VLMs sometimes ignore critical visual details, treating vision as secondary to language. More recent analysis goes further, suggesting that VLMs can be misled by different questioning methods and give incorrect answers to some questions even though it can understand the visual content (Liu et al., 2025; Zhang et al., 2025). These findings raise an intriguing possibility: the problem may not lie solely in "blindness" to images, but rather in what happens after the model has already seen them. Notably, existing studies often attribute this issue to the model's overall lower attention to image tokens compared to text tokens (Liu et al., 2025; Chen et al., 2025), with few studies explore this phenomenon through the lens of the model's internal mechanisms.

In this work, we take a systematic step toward unpacking following intriguing questions for understanding VLMs' visual evidence utilization: (i) How do models balance and transition between textual and visual information across layers? (ii) Which layers are most critical for grounding answers

in the correct evidence? (iii) Can we design interventions to help models actually use what they see? Our analysis reveals several interesting findings:

- **Layer-wise transition.** Shallow layers are text-focused, while deeper layers progressively shift toward images. This sequential process resembles how humans first read the question, then turn their eyes to the picture.

- **Deep-layer visual grounding.** Deep layers do not scatter their attention broadly; instead, they concentrate on localized regions that correspond to key evidence, functioning like a spotlight that cuts through irrelevant clutter.

- **Seeing but not believing.** Perhaps most intriguingly, deep layers often lock onto the correct evidence even when the final answer is wrong. The model *sees*, yet fails to *believe*. This paradox suggests that the bottleneck lies not only in perception but also in how perceived evidence is carried forward into reasoning and generation.

These findings motivate a practical intervention: leveraging deep-layer attention as a signal for guiding models toward more effective use of visual evidence. To this end, we propose an attention-based visual augmentation that highlights the evidence regions attended to by the visual grounding layers, amplifying signals that would otherwise remain buried in its internal representations. This simple yet effective strategy requires no additional training, applies across architectures, and consistently improves answer quality. These results suggest that VLMs already possess latent capabilities for grounding answers in the right evidence, but require targeted elicitation to realize this potential.

In summary, our main contributions are threefold. (i) **Novel Analysis.** We investigate how different attention layers in VLMs process mixed textual and visual inputs, revealing that deep-layer attention reliably identifies the correct evidence regions even when the final answer is incorrect. (ii) **Practical Algorithm.** Building on this insight, we introduce an attention-based augmentation method that highlights the evidence regions identified by the model itself, guiding VLMs to better utilize visual information for factual answering. (iii) **Empirical Study.** We conduct extensive experiments across multiple VLMs and tasks, demonstrating that our method consistently improves answer accuracy and validating both our analysis and the practical effectiveness of attention-based augmentation.

## 2 DISSECTING TEXT AND VISUAL ATTENTION DYNAMICS IN VLMS

To better understand how VLMs process and utilize visual evidence, we conduct a systematic analysis of their internal attention behaviors. Our goal is to uncover not only whether VLMs **perceive** the evidence but also why they fail to **leverage** it for accurate answers. To this end, we organize the section around 4 research questions (RQs): RQ1 analyzes attention transitions between text and image tokens across layers; RQ2 examines layer-specific roles in grounding visual evidence; RQ3 investigates whether models can attend to correct evidence when producing wrong answers; and finally, RQ4 explores reasons for the disconnect between attention to evidence and answer correctness.

### 2.1 RQ1: HOW DOES ATTENTION TRANSITION BETWEEN TEXT AND VISUAL TOKENS?

**Setup.** Recent work has shown that VLMs, on average, assign significantly less attention to visual tokens compared to textual ones (Chen et al., 2025). Here we take a step further by examining how this imbalance evolves across layers. For each layer, we compute the Relative Attention per Token (RAPT), defined as the ratio of section-average attention mass per token to the input-average value.[1] This metric captures how much attention each token type receives on a per-token basis, rather than how attention is distributed as a whole. The results, shown in Figure 1, allow us to trace how the model shifts its focus between linguistic and visual inputs during inference.

**Layer-wise Modality Attention Transition.** Our analysis shows that although image tokens consistently receive less attention per token than text tokens, there is a clear modality shift across depth. Early layers focus overwhelmingly on the question text, reflecting the model's initial parsing of linguistic structure (the apparent crossing of curves in Figure 1 is due to differences in y-axis scaling; text attention values remain larger in magnitude than image attention throughout). As depth increases, however, the balance gradually shifts, with deeper layers allocating relatively more attention to image

---

[1]For example, a RAPT of 0.6 means each token in that section receives 60% of the input-average attention. Results shown here are from the LLaVA-1.5-7B model, with similar trends across other models and datasets.

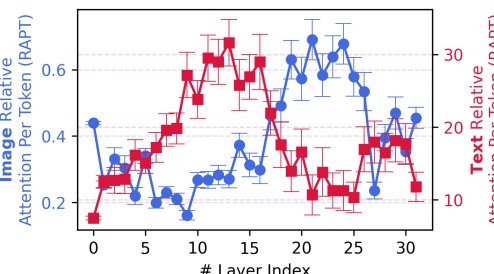

Figure 1: Relative attention per token (RAPT)[1] (y-axis) to text tokens (red) and image tokens (blue) across model layers (x-axis). While early layers strongly emphasize text, deeper layers progressively increase attention to images, showing a sequential transition from linguistic parsing to visual grounding within a single-token inference. *Similar trends hold across different VLM families*, please see Appendix C for more results.

tokens. This layer-wise transition indicates that textual and visual information are not processed in parallel, but rather sequentially, with vision playing a stronger role at later stages of inference.

## 2.2 RQ2: WHICH IMAGE REGIONS DO DIFFERENT LAYERS ATTEND TO?

**Setup.** To further unpack the role of different layers, we analyze not only how much attention is allocated to image tokens, but also *where* this attention is directed. We begin with a visualized case study to intuitively illustrate which regions of the image attract attention across layers. Figure 2 presents three images from VQA pairs and the corresponding attention distributions from different layers. For clarity, we highlight the ground-truth evidence regions with red bounding boxes.

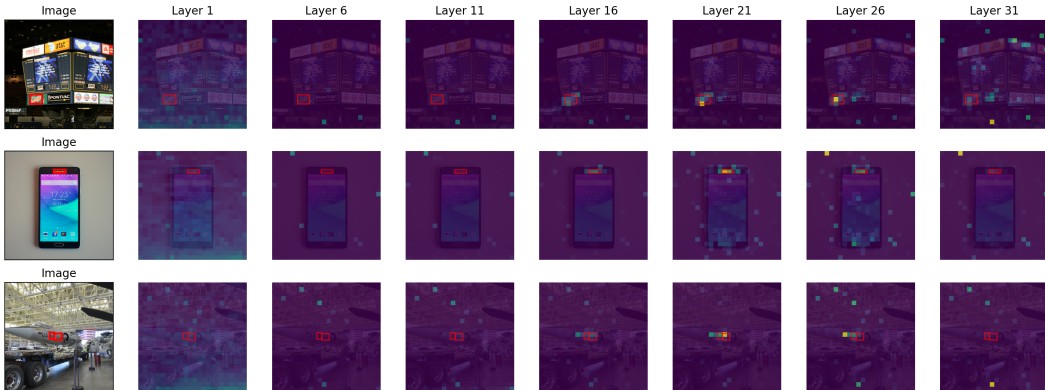

Figure 2: Visualization of image attention across different layers of LLaVA-1.5-7B. Red bounding boxes denote ground-truth evidence regions. While shallow layers exhibit global weak attention, deeper layers consistently focus on localized regions that align with the relevant evidence.

**Deep-layer Visual Grounding.** Our observations show that the first layer distributes attention almost uniformly across all image patches, showing no particular focus. As shown in Fig. 1, subsequent shallow layers (e.g., layers 6–11) allocate little attention to image tokens overall, without exhibiting any distinct spatial patterns. Interestingly, deeper layers (e.g., layers 16–26) display sparse yet highly concentrated attention, consistently highlighting regions aligned with ground-truth evidence[2]. This suggests that deep layers function as visual grounders, filtering irrelevant content and selectively attending to the evidence necessary for answering the question.

## 2.3 RQ3: DO VLMS PERCEIVE VISUAL EVIDENCE WHEN GIVING WRONG ANSWERS?

**Setup.** Building on our findings about the critical role of deep layers in visual perception, a natural follow-up question arises: *are VQA errors always caused by the model's failure to perceive visual evidence?* To answer this, we conduct both **qualitative** and **quantitative** analyses. For the case study, we examine representative VQA samples where the model produces incorrect answers. For the dataset-level study, we leverage the VisualCOT dataset (Shao et al., 2024), which provides human-annotated visual evidence regions, to statistically compare attention to evidence versus non-evidence tokens across layers, conditioned on whether the model's predictions are correct or incorrect.

---

[2]In some images, isolated non-evidence patches receive high attention, consistent with the attention sink phenomenon (Darcet et al., 2024; Kang et al., 2025). This does not affect the overall grounding behavior, and Section 3.2 introduces a sink-token filtering strategy to remove such noise.

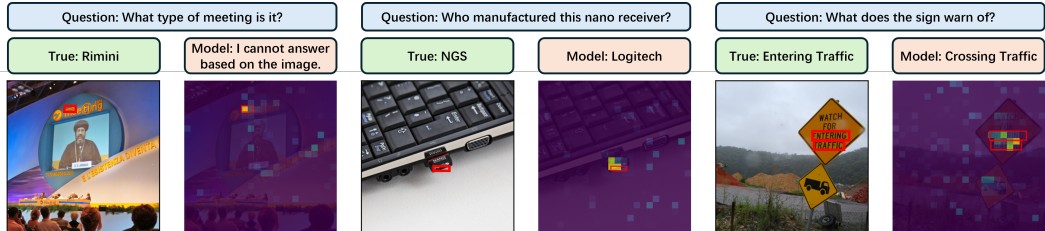

Figure 3: Qualitative examples of **"seeing but not believing"** with LLaVA-1.5-7B model. Each case shows the input image (left) and the average attention map of the late 50% layers (right). Deep layers attend to the correct evidence regions (red boxes), but the final answers are still incorrect due to generation failures. We provide more qualitative examples in Appendix C.7.

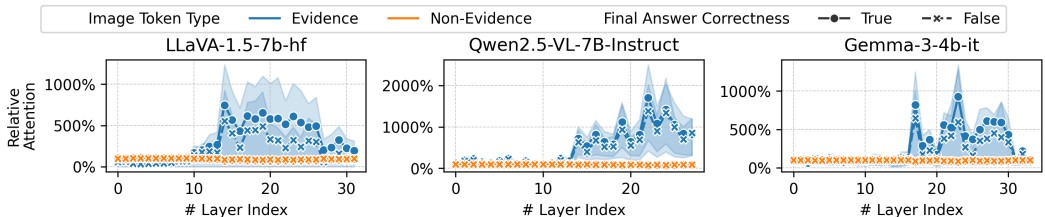

Figure 4: Relative attention to the evidence/non-evidence image tokens (y-axis) across the layers (x-axis) for different VLM families. Deeper layers pay much greater attention to crucial evidence (blue lines) in the context, even when VLM responds incorrectly (dashed lines). Best viewed in color.

**Case Study.** Figure 3 illustrates three typical error cases, including false rejection (refusing to answer despite evidence being present), hallucination (answering with non-existent content), and partially correct responses. In all these examples, the deep layers of the model correctly attend to the relevant evidence regions, yet this perception fails to translate into correct final outputs. We term this phenomenon "seeing but not believing," highlighting that attention to evidence does not necessarily guarantee grounded and correct answers.

**Seeing but Not Believing.** To verify that this phenomenon is not limited to anecdotal cases, we perform a larger statistical analysis using VisualCOT. As shown in Figure 4, deeper layers consistently allocate more attention to ground-truth evidence than to non-evidence tokens. Even when the model produces incorrect answers, its internal attention still concentrates relatively more on the correct evidence regions, albeit with a weaker overall magnitude compared to correct predictions. This "seeing but not believing" phenomenon indicates that while the model often perceives the right visual cues, the subsequent reasoning and generation processes fail to effectively integrate this attended information into the final response.

## 2.4 RQ4: WHY DOES "SEEING BUT NOT BELIEVING" HAPPENS?

The above findings suggest that the visual bottleneck in VLMs is not only perceptual but also cognitive. Although deep layers focus on the correct evidence, models can still fail to translate this perception into factual answers. In this section, we build on our observations and recent literature to discuss two perspectives on the underlying reasons for this counter-intuitive phenomenon.

**Textual Information Dominance.** A growing body of work shows that VLMs frequently exhibit a strong preference for linguistic signals, placing "blind faith in text" even when such signals conflict with visual evidence (Ailin et al., 2025; Kang-il et al., 2024). One explanation points to the architectural imbalance in current VLMs, where a large language model backbone is paired with a relatively small visual encoder, reinforcing the dominance of textual patterns over visual grounding (Shi et al., 2024; Cong et al., 2025; Shengbang et al., 2024). This textual bias has also been linked to multimodal hallucinations: as generation unfolds, the reliance on visual inputs diminishes and language priors increasingly dominate, producing outputs that are linguistically fluent but visually ungrounded (Alessandro et al., 2024; Lanyun et al., 2024; Nanxing et al., 2025). Notably, the second failure case in Figure 3 aligns with this line of evidence: the LLM backbone hallucinates a connection between "logitech" and "nano receiver", likely due to their frequent co-occurrence in the training corpus, while the visual input fails to override this strong textual prior.

**Visual Context Under-utilization.** Another interesting perspective to understand this phenomenon is through studies of context under-utilization in retrieval-augmented generation (RAG), where the image in VQA can be viewed as a form of context. Recent work has shown that LLMs often fail to fully exploit retrieved information, generating incorrect answers even when the necessary facts are present (Garima et al., 2024; Fei et al., 2024). This issue becomes more pronounced when the context contains more irrelevant information (Jirui et al., 2025; Huayang et al., 2024), while emphasizing salient evidence within the context can help models make better use of the provided context (Chang et al., 2024; Mortaheb et al., 2025). In our case, we note that the deep-layer visual grounding behaviors of VLMs can naturally serve as signals for evidence highlighting, guiding the model to attend to critical visual information. This insight directly motivates our method design.

## 3 SIMPLE INFERENCE-TIME VISUAL EVIDENCE AUGMENTATION

Building on the above analysis, we propose an inference-time method that leverages deep-layer visual grounding to address the "seeing but not believing" problem. Given a VQA input, VEA (**V**isual **E**vidence **A**ugmentation) extracts attention from visual grounding layers, applies denoising and smoothing to form a highlighting mask, and overlays it on the original image to create an augmented input. VEA is lightweight, requires no additional training, and its overview is shown in Fig. 5.

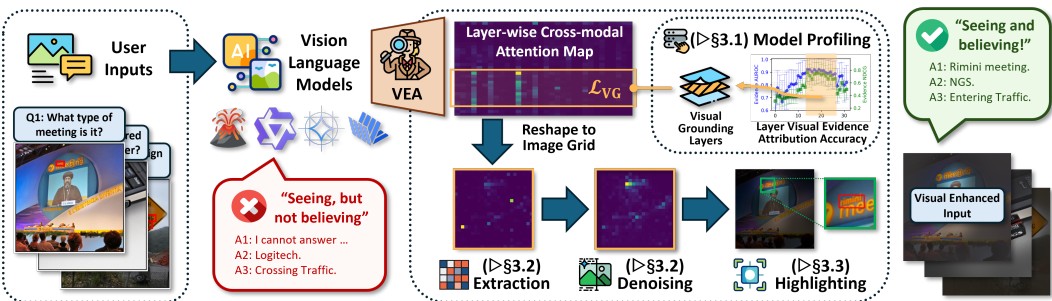

Figure 5: Overview of the proposed VEA framework. Best viewed in color.

### 3.1 VISUAL EVIDENCE ATTRIBUTION LAYER PROFILING

**Notation.** We summarize the key notations of Transformer-based VLMs used in this work, referring readers to Vaswani et al. (2017) for a full exposition and Appendix A.3 for implementation details. Given a VLM $\Phi$, a question $\boldsymbol{q}$, an image $\mathcal{I}$, and a QA prompt $\tau_{\text{QA}}$, the generated answer is $\boldsymbol{g} \leftarrow \Phi(\tau_{\text{QA}}(\mathcal{I}, \boldsymbol{q}))$. A decoder-only Transformer processes an input of $n$ tokens (text and image) and generates each new token by attending to all preceding tokens. For head $h$ in layer $\ell$, the attention is $\boldsymbol{a}^{(\ell,h)} \in \mathbb{R}^n$, and the layer-level vector is $\boldsymbol{a}^{(\ell)} = \frac{1}{H} \sum_{h=1}^{H} \boldsymbol{a}^{(\ell,h)}$, where $H$ is the number of heads. This aggregated vector summarizes how layer $\ell$ distributes attention across input tokens, showing which parts of the input it considers most informative at that stage.

**Layer Attention Extraction and Profiling.** The first step of VEA is to identify Transformer layers with the strongest evidence attribution capability. We use a small diagnostic subset of Visual-COT (Shao et al., 2024), which provides bounding-box annotations of evidence regions. These annotations are aligned with the vision encoder patch space to obtain token-level evidence labels. For each layer, we extract its visual attention vector and compute the AUROC against the ground-truth labels, measuring the attribution quality of that layer. Formally, let an image $\mathcal{I}$ be divided into $m$ patches $\mathcal{P}_{\mathcal{I}} = \{\boldsymbol{p}_1, \dots, \boldsymbol{p}_m\}$, each mapped to a visual token. A binary label vector $\boldsymbol{y}_{\mathcal{I}} = [y_1, \dots, y_m]^{\top} \in \{0, 1\}^m$ is defined by setting $y_i = 1$ if patch $\boldsymbol{p}_i$ overlaps with any annotated evidence region. Denote $i_{\mathcal{I}}^{\text{start}}$ as the index of the first visual token in $\tau(\mathcal{I}, \boldsymbol{q})$. The patch-level visual attention vector of layer $\ell$ is then $\bar{\mathbf{a}}_{\mathcal{I}}^{(\ell)} = [a_{i_{\mathcal{I}}^{\text{start}}}^{(\ell)}, \dots, a_{i_{\mathcal{I}}^{\text{start}}+m-1}^{(\ell)}] \in \mathbb{R}^m$. We compute $\text{AUROC}(\boldsymbol{y}_{\mathcal{I}}, \bar{\mathbf{a}}_{\mathcal{I}}^{(\ell)})$ as the attribution score of layer $\ell$ for $\mathcal{I}$. Finally, we select the top 10% of layers with the highest average scores[3] as the set of visual-grounding layers $\mathcal{L}_{\text{VG}}$, which will be used for subsequent inference-time evidence attribution. We note that this profiling is a one-time, model-level cost and does not need to be repeated for each question at inference time. More details are in Appendix A.3.

---

[3]We note that the diagnostic set does not need to be large, ∼100 examples is sufficient to obtain stable results.

## 3.2 Inference-time Visual Evidence Attribution

**Attention Extraction.** At inference time, VEA leverages the VLM's internal attention representations to highlight the most relevant evidence regions in the image. We note that, as shown in the analysis section, the evidence-reading layer's visual grounding pattern appears when generating the first answer token. Therefore, at this attention extraction step, we can perform only an efficient single-token forward pass rather than generating the full answer. Given the set of visual-grounding layers $\mathcal{L}_{\text{VG}}$ identified in the profiling stage, we compute for each image $\mathcal{I}$ a patch evidence score vector $\boldsymbol{e}_{\mathcal{I}} \in \mathbb{R}^m$ over its patch set $\mathcal{P}_{\mathcal{I}} = \{\boldsymbol{p}_1, \ldots, \boldsymbol{p}_m\}$. For layer $\ell \in \mathcal{L}_{\text{VG}}$, let $\bar{\mathbf{a}}_{\mathcal{I}}^{(\ell)} = [\bar{a}_1^{(\ell)}, \ldots, \bar{a}_m^{(\ell)}] \in \mathbb{R}^m$ denote its normalized patch-level visual attention vector. The evidence score $e_i$ for patch $\boldsymbol{p}_i$ is then defined as the average attention weight assigned by layers in $\mathcal{L}_{\text{VG}}$:

$$\boldsymbol{e}_{\mathcal{I}} = [e_1, e_2, \ldots, e_m]^\top, \quad e_i = \frac{1}{|\mathcal{L}_{\text{VG}}|} \sum_{\ell \in \mathcal{L}_{\text{VG}}} \bar{a}_i^{(\ell)}, \quad i = 1, \ldots, m. \tag{1}$$

**Attention Mask Denoising.** We reshape the patch evidence score vector $\boldsymbol{e}_{\mathcal{I}} \in \mathbb{R}^m$ into a two-dimensional grid $\boldsymbol{e} \in \mathbb{R}^{H \times W}$ to obtain a 2-D visual evidence mask. However, we note that (also can be observed in Fig. 2 and 3) the raw mask often contains spurious high-valued patches scattered in regions unrelated to the true evidence. Those high-attention patches stem from the attention sink phenomenon (Kang et al., 2025; Darcet et al., 2024): certain visual tokens exhibit abnormally large activations in specific hidden-state dimensions inherited from the language backbone, which causes them to attract disproportionate attention despite carrying little or no semantic content. Unlike genuine evidence regions that form spatially coherent clusters tied to the query, these sink-driven patches appear as isolated artifacts. To suppress those sink tokens, we apply an additional neighborhood-based filtering step. Let $\boldsymbol{e}_{i,j}$ denote the score of the patch at grid location $(i, j)$, and let $\mathcal{N}(i, j)$ denote its $3 \times 3$ neighborhood excluding $(i, j)$. We update $\boldsymbol{e}_{i,j}$ by comparing it against its neighbors:

$$\boldsymbol{e}_{i,j}' = \begin{cases} \frac{1}{|\mathcal{N}(i,j)|} \sum_{(p,q) \in \mathcal{N}(i,j)} \boldsymbol{e}_{p,q}, & \text{if } \boldsymbol{e}_{i,j} > \lambda \cdot \max_{(p,q) \in \mathcal{N}(i,j)} \boldsymbol{e}_{p,q}, \\ \boldsymbol{e}_{i,j}, & \text{otherwise,} \end{cases} \tag{2}$$

where $\lambda$ is a multiplicative threshold controlling the strictness of noise suppression. We set $\lambda = 10$ in our experiment, i.e., a patch is regarded as noise and replaced by the local average if its score is more than one order of magnitude larger than all of its neighbors. This denoising step yields a cleaner and more spatially coherent evidence mask, which better highlights the true evidence regions.

## 3.3 Attention-Guided Visual Evidence Highlighting

**Highlight Mask Smoothing.** While neighborhood filtering suppresses isolated outliers, the resulting token-level mask $\boldsymbol{e}'$ often introduces unnatural, mosaic-like artifacts when applied to the pixel-level image, leading to sharp local fluctuations that may hinder the VLM's understanding of image content. To further enhance spatial coherence and ensure that contiguous evidence regions form smooth clusters, we apply a Gaussian smoothing step. This operation distributes attention scores more evenly within local neighborhoods, reducing noise while preserving the overall evidence structure. Formally, let $G_\sigma \in \mathbb{R}^{k \times k}$ be a Gaussian kernel with standard deviation $\sigma$. The smoothed mask $\tilde{\boldsymbol{e}}$ is obtained by convolving $\boldsymbol{e}'$ with $G_\sigma$:

$$\tilde{\boldsymbol{e}}_{i,j} = \sum_{p=-r}^{r} \sum_{q=-r}^{r} G_\sigma(p, q) \, \boldsymbol{e}_{i-p, j-q}', \tag{3}$$

where $r = \lfloor k/2 \rfloor$ and $\sum_{p,q} G_\sigma(p, q) = 1$. In our implementation, the effective $\sigma$ is set by multiplying a hyperparameter *smooth strength* $\sigma \in [0, 1]$ with the shorter side of the image in pixels, ensuring that smoothing naturally adapts to different resolutions. Detailed experiments and discussions are provided in Section 4.2, validating the benefit of adaptive smoothing.

**Visual Evidence Highlighting.** Given the refined mask $\tilde{\boldsymbol{e}} \in \mathbb{R}^{H \times W}$, we next guide the model by directly emphasizing evidence regions. We blend the mask with the original RGB image $\mathcal{I} \in \mathbb{R}^{H \times W \times 3}$ so that high-attention areas are preserved and low-attention areas visually down-weighted. This emphasizes critical evidence while reducing irrelevant content. Concretely, the augmented image $\hat{\mathcal{I}}$ is constructed by attenuating low-attention regions, controlled by $\alpha \in [0, 1]$:

$$\hat{\mathcal{I}}_{i,j,c} = \big(\alpha + (1 - \alpha)\tilde{\boldsymbol{e}}_{i,j}\big) \mathcal{I}_{i,j,c}, \tag{4}$$

for each pixel $(i, j)$ and channel $c$. Large $\tilde{e}_{i,j}$ (evidence) keeps pixel values close to the original, while small $\tilde{e}_{i,j}$ (non-evidence) darkens the region toward $\alpha$. This simple augmentation makes evidence more salient without retraining, steering inference toward highlighted regions. We validate the effect of different highlight strength choices in Section 4.2. This concludes the description of VEA, we next evaluate its effectiveness across diverse VLMs and tasks.

## 4 EXPERIMENT RESULTS AND ANALYSIS

In this section, we conduct systematic experiments across eight VLMs from four different families with varying sizes and on four visual evidence retrieval-based VQA tasks from diverse domains to examine the following research questions (RQs): **RQ1:** How does VEA perform in improving VLM response quality and accuracy? **RQ2:** To what extent do the visual evidence regions highlighted by VEA align with the ground-truth visual evidence? **RQ3:** How robust is VEA under different types of image noise perturbations? **RQ4:** How do different parameter choices and the inclusion or removal of specific modules influence the performance of VEA?

### 4.1 EXPERIMENTAL SETUP

**Datasets and Metrics.** The primary goal of our experiments is to test whether VEA and related baselines help VLMs attend to fine-grained visual evidence for more accurate responses. For this purpose, we adopt four evidence-based VQA datasets from the VisualCoT benchmark (Shao et al., 2024): InfoVQA (Mathew et al., 2022), DocVQA (Mathew et al., 2021), SROIE (Huang et al., 2019), and TextVQA (Singh et al., 2019). These tasks require models to extract localized information from images, such as text snippets in natural scenes or documents, and provide human-annotated evidence that allows us to directly measure both QA accuracy and the quality of visual grounding. We use greedy decoding for deterministic outputs, reporting Exact Match and Token-level F1 as QA metrics, and AUROC and NDCG@all as evidence attribution metrics. Please see more details in Appendix A.

**Models and Baselines.** We evaluate the effectiveness and generality of VEA using the most recent series from four popular VLM families: LLaVA-Next (Liu et al., 2024), Qwen2.5VL (Bai et al., 2025), Gemma3 (Team et al., 2025), and InternVL3.5 (Wang et al., 2025). We note that LLaVA-Next and InternVL3.5 exhibit memory leakage when extracting attention, so we use Qwen2.5VL as a delegate model for image augmentation. For comparison, we also include several inference-time text/image augmentation baselines designed to enhance visual information utilization. (i) **Instructioning** is a simple baseline for evaluating whether prompting alone is sufficient; it explicitly instructs the model to attend to visual evidence. (ii) **CGR** (Liu et al., 2025) employs a two-step reasoning process in which the model first extracts detailed information from the image and then generates answers from this intermediate representation. (iii) **VAR** (Liu et al., 2025) uses attention scores from the final layer to apply a binary mask that highlights salient regions. (iv) **AGLA** (An et al., 2025) adopts a GradCAM-based (Selvaraju et al., 2017) approach that masks irrelevant regions and ensembles outputs from the original and masked images to strengthen visual grounding.

### 4.2 RESULTS AND ANALYSIS

**RQ1: VEA consistently improves VLM's visual grounding.** Table 1 reports the average performance of different visual evidence augmentation methods across 8 VLMs from 4 latest model families. We observe that all augmentation methods yield improvements over the BASE models, confirming the effectiveness of explicitly enhancing visual grounding. Among the baselines, CGR, VAR, and AGLA consistently outperform simple prompting (INST), with AGLA showing the strongest gains overall. Nevertheless, our proposed VEA achieves the best results across both metrics and all model families. Specifically, it delivers an average improvement of $+5.67$ points (up to $+11.1$) in Exact Match and over $+6.83$ points (up to $+17.3$) in Token F1 compared to the base models, while also achieving the lowest average rank (1.12 and 1.22) among all baselines. The gains of VEA are especially pronounced on smaller-scale models (e.g., LLaVA-7B and Gemma-4B), suggesting that VEA is particularly effective in compensating for the weaker visual grounding capability of smaller VLMs, while still providing consistent benefits for larger models. These results demonstrate that VEA provides more robust and consistent gains across diverse VLM architectures and scales, establishing it as a strong and generalizable approach for improving visual evidence utilization.

**RQ2: VEA accurately highlight visual evidence.** To validate the quality of VEA evidence attribution, we evaluate how well the extracted evidence aligns with human annotation. Following Section 3.1, where we compute token-level ground-truth evidence label vectors $\boldsymbol{y}_{\mathcal{I}} \in \{0, 1\}^m$ and

Table 1: Main results of applying visual evidence augmentation methods to 8 VLMs from 4 latest series of LLaVA, Qwen, Gemma, and InternVL families. Due to space limitation, we report the averaged results on 4 visual question answering tasks. For each metric, we also report the average rank of each method over all tested tasks and models. *Detailed results can be found in Appendix C.*

| Metric | Method | LLaVA-NeXT 7B | 13B | Qwen2.5-VL 7B | 32B | Gemma3 4B | 27B | InternVL3.5 8B | 14B | Avg. Rank ($\downarrow$) |
|---|---|---|---|---|---|---|---|---|---|---|
| **Exact Match** | BASE | 38.5 | 49.4 | 73.4 | 69.3 | 56.6 | 69.3 | 79.3 | 79.3 | 5.38 |
| | INST | 38.8(+0.4) | 50.2(+0.7) | 73.9(+0.4) | 69.0(-0.3) | 56.0(-0.6) | 70.2(+0.8) | 79.2(-0.1) | 78.8(-0.5) | 5.47 |
| | CGR | 45.4(+7.0) | 52.7(+3.3) | 76.1(+2.6) | 73.0(+3.7) | 60.4(+3.7) | 72.3(+2.9) | 82.4(+3.1) | 81.7(+2.4) | 3.09 |
| | VAR | 42.7(+4.2) | 51.5(+2.1) | 76.4(+3.0) | 70.4(+1.1) | 58.0(+1.3) | 73.4(+4.1) | 80.2(+0.9) | 80.4(+1.1) | 3.44 |
| | AGLA | 46.4(+7.9) | 53.4(+4.0) | 77.9(+4.4) | 73.8(+4.4) | 60.4(+3.8) | 74.2(+4.9) | 82.3(+3.0) | 82.3(+3.0) | 2.50 |
| | VEA | **49.6**(+11.1) | **54.1**(+4.7) | **78.4**(+4.9) | **75.8**(+6.5) | **61.2**(+4.6) | **75.3**(+6.0) | **83.2**(+3.9) | **82.9**(+3.6) | **1.12** |
| **Token F1** | BASE | 33.3 | 53.8 | 77.7 | 69.8 | 50.5 | 57.7 | 79.6 | 69.4 | 5.53 |
| | INST | 38.4(+5.1) | 54.4(+0.7) | 77.6(-0.1) | 71.2(+1.4) | 51.9(+1.4) | 58.3(+0.6) | 80.0(+0.4) | 69.0(-0.4) | 5.28 |
| | CGR | 41.0(+7.7) | 56.8(+3.0) | 80.9(+3.2) | 74.8(+5.0) | 54.7(+4.2) | 61.3(+3.7) | 84.5(+4.9) | 70.5(+1.1) | 3.44 |
| | VAR | 45.7(+12.4) | 55.7(+2.0) | 79.0(+1.3) | 73.9(+4.1) | 56.4(+5.9) | 61.3(+3.6) | 84.2(+4.6) | 70.3(+0.9) | 3.22 |
| | AGLA | 45.1(+11.8) | 58.7(+5.0) | 81.9(+4.2) | 75.2(+5.3) | 55.9(+5.4) | 61.4(+3.8) | 85.2(+5.6) | 70.8(+1.4) | 2.31 |
| | VEA | **50.6**(+17.3) | **59.2**(+5.4) | **82.1**(+4.4) | **76.9**(+7.1) | **57.8**(+7.3) | **62.3**(+4.6) | **85.8**(+6.2) | **71.7**(+2.3) | **1.22** |

Table 2: Comparison of visual evidence attribution accuracy (token-level AUROC and NDCG@all) of different attribution methods. Methods denoted by two percentages are static baselines that use fixed span of layers to compute visual token evidence scores (e.g., $L_{50\%-100\%}$ uses the late 50% of layers as visual grounding layers). Similar to Table 1, we report the average results of 4 tasks due to space limitation, *full detailed results can be found in Appendix C.*

| Evidence Attribution | LLaVA 7B AUROC | NDCG | LLaVA 13B AUROC | NDCG | Qwen 7B AUROC | NDCG | Qwen 32B AUROC | NDCG | Gemma 4B AUROC | NDCG | Gemma 27B AUROC | NDCG | Avg. Rank ($\downarrow$) AUROC | NDCG |
|---|---|---|---|---|---|---|---|---|---|---|---|---|---|---|
| $L_{0\%-100\%}$ | 75.9 | 47.2 | 76.3 | 47.2 | 68.5 | 41.7 | 57.0 | 33.0 | 59.5 | 35.5 | 61.8 | 36.4 | 4.33 | 4.42 |
| $L_{0\%-50\%}$ | 68.2 | 43.2 | 73.1 | 46.4 | 59.4 | 34.2 | 51.3 | 31.9 | 56.5 | 34.3 | 55.4 | 34.9 | 5.67 | 5.67 |
| $L_{50\%-100\%}$ | 78.0 | 54.5 | 76.9 | 50.5 | 79.5 | 58.1 | 67.6 | 43.3 | 65.9 | 43.7 | 68.1 | 43.6 | 2.88 | 2.83 |
| VAR | 70.8 | 45.1 | 72.1 | 44.0 | 75.2 | 54.1 | 65.7 | 39.8 | 51.2 | 33.3 | 58.2 | 36.8 | 4.92 | 4.88 |
| AGLA | 80.2 | 57.2 | 81.1 | 56.6 | 77.7 | 55.4 | 75.1 | 51.3 | 68.3 | 44.5 | 73.8 | 49.0 | 2.21 | 2.21 |
| VEA | **83.6** | **63.5** | **84.4** | **63.5** | **85.2** | **68.6** | **79.1** | **58.4** | **80.0** | **59.9** | **81.2** | **60.1** | **1.00** | **1.00** |

measure their alignment with visual attention vectors $e_{\mathcal{I}} \in \mathbb{R}^m$ using AUROC and NDCG. Table 2 compares different attribution strategies. Static baselines that aggregate attention from fixed spans of layers show that later layers generally provide stronger visual grounding than earlier ones, as evidenced by the higher scores of $L_{50\%-100\%}$ over $L_{0\%-50\%}$. However, such static choices are still suboptimal compared to adaptive approaches. Among adaptive methods, AGLA outperforms VAR, but both are consistently outperformed by our proposed VEA, which achieves the best performance across all model families and metrics. Overall, the absolute accuracies are relatively high (e.g., AUROC often exceeds 80 and NDCG surpasses 60), suggesting that the internal attention signals of VLMs are capable of localizing visual evidence with reasonable accuracy.

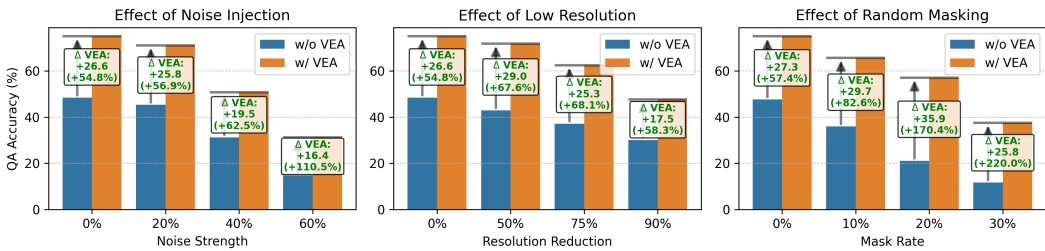

Figure 6: Robustness of VEA against various types of visual perturbations.

**RQ3: VEA is robust to visual noise and corruptions.** In real-world applications, visual inputs are often corrupted by noise, low resolution, or occlusion, making robustness a critical requirement for VLMs. To validate the robustness of our method, we evaluate the effect of visual evidence augmentation under three perturbation settings. (i) **Noise Injection:** additive Gaussian noise with varying strength, where 100% corresponds to pure noise. (ii) **Low Resolution:** downsampling the image by number of pixels, e.g., a 90% reduction results in 10% of the original pixels. (iii) **Random**

**Masking:** randomly masking out $x\%$ of visual patches. We report the exact match accuracy of LLaVA-NeXT-7B on the TextVQA dataset in Figure 6. The results show that the base model degrades sharply, while VEA consistently improves robustness across all types and levels of perturbations. Even under extreme conditions such as $60\%$ noise or $30\%$ random masking, it still delivers improvements of $+16.4$ and $+25.8$ points, corresponding to relative gains of over $110\%$ and $220\%$. These results indicate that the evidence-driven signals elicited by VEA provide strong guidance, enabling the model to maintain accurate reasoning even when raw visual inputs are severely degraded.

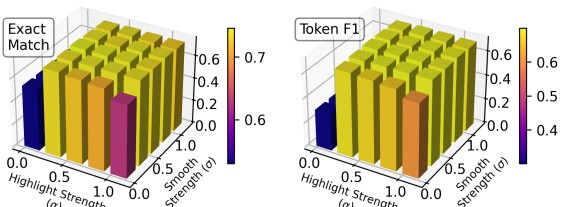

Figure 7: Parameter analysis of VEA.

Table 3: Ablation study results.

| Setting | Exact Match | Token F1 |
|---|---|---|
| VEA | 73.4 | 68.1 |
| w/o Denoise | 70.9 (-2.52) | 64.9 (-3.12) |
| w/o Profiling | 71.0 (-2.42) | 65.3 (-2.78) |
| w/o Smoothing | 68.3 (-5.12) | 62.8 (-5.27) |

**RQ4: Effect of different parameter choice.** We analyze two parameters of VEA: the *highlight strength* $\alpha$, which controls how strongly non-evidence regions are suppressed (larger values yield darker backgrounds), and the *smooth strength* $\sigma$, which determines the degree of spatial smoothing applied to the evidence mask. To ensure consistent smoothing across images of different resolutions, the effective kernel size is set as $\sigma$ multiplied by the shorter image side. As shown in Figure 7, we observe that: (i) our method is overall robust to a wide range of parameter choices; (ii) overly strong highlighting removes too much visual context, while the absence of smoothing also hinders VLMs' understanding of the image. Balancing these factors, we set both $\alpha$ and $\sigma$ to 0.5 in all experiments.

**RQ4: Ablation Study** Table 3 reports the effect of removing different components from VEA. We find that each component contributes positively to the final performance: removing the denoising step or the profiling step results in moderate drops (about $-2.5$ EM and $-3.0$ F1), while removing the smoothing step leads to the largest degradation ($-5.1$ EM and $-5.3$ F1). These results confirm that denoising, profiling, and smoothing are all necessary for maximizing the effectiveness of VEA.

**More Results and Analysis.** Due to space limitations, we provide summarized results in the main text to highlight key analyses and insights. Please refer to Appendix A for reproducibility details, Appendix B for discussions on limitations and future directions in applying our findings in VLM reasoning, and Appendix C for complete results and additional analyses, including per-model and per-dataset QA performance, evidence attribution accuracy, and layer-wise attention dynamics, etc.

## 5 RELATED WORKS

**Challenges in Visual Evidence Utilization.** Despite significant progress on multimodal tasks such as VQA (Singh et al., 2019; Mathew et al., 2021; Huang et al., 2019; Mathew et al., 2022), VLMs often fail to fully leverage visual evidence. Studies report a strong tendency to rely on linguistic priors even when they conflict with the image (Ailin et al., 2025; Kang-il et al., 2024; Shengbang et al., 2024), leading to hallucinations and visually inconsistent outputs (Alessandro et al., 2024; Lanyun et al., 2024; Nanxing et al., 2025; Cong et al., 2025). This issue is further amplified by the imbalance between the language backbone and the vision encoder (Shi et al., 2024; Shengbang et al., 2024). Similar challenges arise in retrieval augmented generation, where models often under use retrieved evidence (Garima et al., 2024; Hexiang et al., 2024; Evgenii et al., 2024; Fei et al., 2024). Recent efforts have started addressing this broader problem by refining the visual input itself through model internal signals, including attention guided cropping (Zhang et al., 2025) and mask generation (Liu et al., 2025; An et al., 2025), illustrating the potential of training free visual enhancement for improving evidence utilization.

**Attention and Interpretability in VLMs.** Attention patterns offer an informative view of multimodal processing. Prior analyses show that shallow layers focus primarily on textual tokens while deeper layers begin to attend to localized image regions, sometimes aligning with ground truth evidence even when predictions remain incorrect (Liu et al., 2025; Chen et al., 2025; Tong et al., 2024). These observations motivate a wide spectrum of interpretability and intervention techniques, including GradCAM style attribution (Selvaraju et al., 2017; An et al., 2025), masking based strategies (Liu

et al., 2025), and inference time attention modification to mitigate hallucination (Shi et al., 2024; Alessandro et al., 2024). More work ViCrop (Zhang et al., 2025) follows this direction by using layer specific attention to identify a better crop that enhances perception. Our work extends this direction by demonstrating that aggregated deep-layer attention, once denoised and smoothed, provides a stable training-free signal for inference-time evidence highlighting, connecting analysis with practical improvements in visual grounding.

## 6  CONCLUSIONS

In this work, we systematically investigated how vision-language models (VLMs) attend to textual and visual information, and uncovered a key disconnect between visual perception and answer correctness. Our analysis revealed three important findings: attention transitions from text to image across layers, deep layers act as visual grounders by sparsely focusing on evidence regions, and models often "seeing but not believing", perceiving the right evidence but failing to use it for correct answers. Building on these insights, we introduced VEA, an inference-time method that highlights deep-layer evidence and consistently improves factual answering across models and benchmarks without additional training. Our results suggest that internal attention patterns already encode reliable visual cues, and making them explicit can help bridge perception and reasoning.

## ETHICS STATEMENT

While our findings reveal potential weaknesses in VLMs' reasoning and grounding, we focus on diagnostic analysis and training-free interventions rather than deployment-ready systems, minimizing risks of harmful misuse. Like other research focused on vision language models, our work has potential social impacts, but none of which we feel must be highlighted here. All authors have reviewed the ICLR Code of Ethics and affirm adherence to its principles throughout this work.

## REPRODUCIBILITY STATEMENT

We made extensive efforts to ensure the reproducibility of our results. The experimental setup, including datasets, evaluation metrics, and implementation of our method and baselines, is fully described in Section 4.1 and Appendix A. Specifically, Appendix A.1 details dataset processing and evaluation metrics, Appendix A.2 lists all models and their Hugging Face checkpoints, and Appendix A.3 provides implementation details of VEA and baselines. Complete per-model and per-dataset results, evidence attribution scores, and layer-wise attention dynamics are provided in Appendix C.

### ACKNOWLEDGMENTS

This work is supported by NSF (2416070). The content of the information in this document does not necessarily reflect the position or the policy of the Government, and no official endorsement should be inferred. The U.S. Government is authorized to reproduce and distribute reprints for Government purposes notwithstanding any copyright notation here on.

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

APPENDIX

## A  REPRODUCIBILITY DETAILS

### A.1  DATASETS AND METRICS

**Datasets.** The primary goal of our experiments is to test whether VEA and related baselines enable VLMs to attend to and exploit fine-grained visual evidence for more accurate responses. To this end, the datasets should evaluate both a model's ability to *extract localized information from images* during answer generation and its accuracy in grounding answers to visual evidence. We adopt four evidence-based VQA datasets from the VisualCoT benchmark (Shao et al., 2024): InfoVQA (Mathew et al., 2022), DocVQA (Mathew et al., 2021), SROIE (Huang et al., 2019), and TextVQA (Singh et al., 2019). These tasks require models to identify fine-grained content, such as text snippets in natural scenes or documents, which directly supports our objective. Importantly, VisualCoT provides pixel-level evidence annotations in the form of bounding boxes for each VQA pair, enabling quantitative evaluation of evidence attribution in addition to standard QA accuracy.

**Metrics.** We use greedy decoding for deterministic outputs. QA performance is measured by *Exact Match (EM)* and *Token-level F1*. Let $\hat{A}$ denote the set of predicted tokens and $A$ the ground-truth tokens:

$$\text{EM}(\hat{A}, A) = \mathbb{I}(\hat{A} = A), \quad \text{F1}(\hat{A}, A) = \frac{2 \cdot |\hat{A} \cap A|}{|\hat{A}| + |A|}.$$

For evidence attribution evaluation, we first align the bounding-box annotations with the vision encoder patch space. Given an image $\mathcal{I}$ divided into $m$ patches $\mathcal{P}_{\mathcal{I}} = \{\boldsymbol{p}_1, \ldots, \boldsymbol{p}_m\}$, we define the binary ground-truth evidence label vector as

$$\boldsymbol{y}_{\mathcal{I}} = [y_1, \ldots, y_m]^\top \in \{0, 1\}^m,$$

where $y_i = 1$ if patch $\boldsymbol{p}_i$ overlaps with any annotated evidence region and $y_i = 0$ otherwise. With predicted evidence scores $\hat{\boldsymbol{p}}_{\mathcal{I}} = [\hat{p}_1, \ldots, \hat{p}_m]^\top \in [0, 1]^m$, *AUROC* is

$$\text{AUROC}(\boldsymbol{y}_{\mathcal{I}}, \hat{\boldsymbol{p}}_{\mathcal{I}}) = \frac{1}{|\mathcal{P}^+||\mathcal{P}^-|} \sum_{i \in \mathcal{P}^+} \sum_{j \in \mathcal{P}^-} \mathbb{I}(\hat{p}_i > \hat{p}_j),$$

where $\mathcal{P}^+ = \{i \mid y_i = 1\}$ and $\mathcal{P}^- = \{j \mid y_j = 0\}$. *NDCG@all* evaluates ranking quality:

$$\text{DCG@all} = \sum_{i=1}^{m} \frac{2^{y_{\pi(i)}} - 1}{\log_2(i+1)}, \quad \text{NDCG@all} = \frac{\text{DCG@all}}{\text{IDCG@all}},$$

where $\pi(i)$ is the index of the $i$-th ranked patch according to $\hat{p}_i$, and IDCG@all is the DCG of the ideal ranking. Higher EM, F1, AUROC, and NDCG@all indicate better QA performance and stronger evidence attribution.

## A.2 MODELS AND BASELINES

We evaluate the effectiveness and generality of VEA using recent models from four popular VLM families: LLaVA-Next (Liu et al., 2024), Qwen2.5VL (Bai et al., 2025), Gemma3 (Team et al., 2025), and InternVL3.5 (Wang et al., 2025). For reproducibility, we list all models with their publicly available checkpoints on Hugging Face[4]. All experiments are conducted on a single NVIDIA A100 GPU with 80GB memory, implemented using the `transformers` and `PyTorch` libraries under `bfloat16` mixed precision. We note that both LLaVA-Next and InternVL3.5 exhibit memory leakage when extracting raw attention maps during inference. Specifically, when using the configuration `attn_implementation='eager'` to obtain layer-wise attention outputs, even the 7B/8B models in these families cause out-of-memory errors on an 80GB GPU under `bfloat16` precision. To address this issue, we employ Qwen2.5VL-7B as a delegate model for VEA. This choice is motivated by its relatively efficient inference speed and stable evidence attribution quality. Concretely, we run VEA on Qwen2.5VL-7B to perform visual evidence attribution and generate augmented images, which are then fed into the original models for answer generation.

For baselines, we include several inference-time augmentation methods that aim to improve visual information utilization without additional training. INST is a prompting baseline that explicitly directs the model to focus on visual evidence, testing whether prompting alone suffices. CGR (Liu et al., 2025) employs a two-step reasoning process where the model first extracts detailed information from the image and then generates answers based on this intermediate representation. VAR (Liu et al., 2025) applies a binary mask derived from final-layer attention scores to highlight salient regions. Finally, AGLA (An et al., 2025) adopts a GradCAM-based (Selvaraju et al., 2017) approach that masks irrelevant regions and ensembles outputs from the original and masked images to reinforce visual grounding.

## A.3 VEA AND BASELINES IMPLEMENTATION DETAILS

Given a VLM $\Phi$, question $\boldsymbol{q}$, image $\mathcal{I}$, and a QA prompt template $\tau_{\text{QA}}$, we obtain the generated answer $\boldsymbol{g}$ for the question by combining the image and question as input: $\boldsymbol{g} \leftarrow \Phi(\tau_{\text{QA}}(\mathcal{I}, \boldsymbol{q}))$. We use the following template as the base prompt for VQA:

> **Base prompt template $\tau_{\text{QA}}$ for image-based VQA**
>
> Directly answer the question based on the image, no explanation is needed. If the image does not contain any relevant evidence, output "I cannot answer based on the given image." Image: `{image}` Question: `{question}`

**VEA Implementation Details.** We first input the image–question pair into the model with the base prompt $\tau_{\text{QA}}$ and extract its attention maps over the input sequence. At this stage, we perform only a single-token forward pass rather than generating the full answer, making the process more efficient than captioning-based approaches such as CGR (Liu et al., 2025). As described in Section 3.1, we then conduct a profiling step to identify the Transformer layers with the strongest evidence attribution

---

[4]LLaVA-1.5-7B: https://huggingface.co/llava-hf/llava-1.5-7b-hf
LLaVA-1.5-13B: https://huggingface.co/llava-hf/llava-1.5-13b-hf
LLaVA-1.6-Mistral-7B: https://huggingface.co/llava-hf/llava-v1.6-mistral-7b-hf
LLaVA-1.6-Vicuna-13B: https://huggingface.co/llava-hf/llava-v1.6-vicuna-13b-hf
Qwen2.5-VL-7B: https://huggingface.co/Qwen/Qwen2.5-VL-7B-Instruct
Qwen2.5-VL-32B: https://huggingface.co/Qwen/Qwen2.5-VL-32B-Instruct
Gemma-3-4B: https://huggingface.co/google/gemma-3-4b-it
Gemma-3-27B: https://huggingface.co/google/gemma-3-27b-it
InternVL3.5-8B: https://huggingface.co/OpenGVLab/InternVL3_5-8B-HF
InternVL3.5-14B: https://huggingface.co/OpenGVLab/InternVL3_5-14B-HF

capability. Specifically, we use 100 examples from the TextVQA dataset as a diagnostic subset, compute the AUROC between the patch-level attention scores and ground-truth evidence labels, and select the top 10% of layers (rounded up) with the highest average AUROC scores. Table 4 reports the average attribution quality of all layers and the subset of selected visual-grounding layers $\mathcal{L}_{\text{VG}}$ for representative models.

Table 4: Profiling results of visual-grounding layers $\mathcal{L}_{\text{VG}}$ for representative models. We report the total number of layers, the average AUROC across all layers, and the statistics of the selected top layers with the strongest evidence attribution capability.

| Model ID | Full Layers | | Visual Grouding Layers ($\mathcal{L}_{\text{VG}}$) | | |
|---|---|---|---|---|---|
| | #Layers | Avg. AUROC | #Layers | Avg. AUROC | Layer IDs |
| llava-hf/llava-1.5-7b-hf | 32 | 83.98 | 4 | 92.13 | {14,15,17,19} |
| llava-hf/llava-1.5-13b-hf | 40 | 85.96 | 4 | 92.16 | {13,14,15,16} |
| Qwen/Qwen2.5-VL-7B-Instruct | 28 | 80.07 | 3 | 89.09 | {18,22,24} |
| Qwen/Qwen2.5-VL-32B-Instruct | 64 | 73.08 | 7 | 88.18 | {49,50,51,52,53,55,56} |
| google/gemma-3-4b-it | 34 | 65.40 | 4 | 80.32 | {17,19,21,23} |
| google/gemma-3-27b-it | 62 | 68.97 | 7 | 84.70 | {35,37,40,41,47,53,58} |

Once the set $\mathcal{L}_{\text{VG}}$ is identified, we aggregate attention signals from these layers to construct the visual evidence attribution map. To improve robustness, we apply denoising and smoothing operations with hyperparameters highlight strength $\alpha = 0.5$ and smooth strength $\sigma = 0.5$, following the discussion in Section 3.3 and Section 4.2. The resulting attention map is then overlaid onto the input image, highlighting the most relevant regions. Finally, we construct an augmented prompt $\tau_{\text{VEA}}$ that explicitly instructs the model to focus on highlighted evidence regions when generating answers.

> **Prompt template $\tau_{\text{VEA}}$ for VQA with VEA-augmented Images**
>
> Directly answer the question based on the image, no explanation is needed. If the image does not contain any relevant evidence, output "I cannot answer based on the given image." Only use words from the picture, especially those in the highlighted region, to answer the question. Image: {image} Question: {question}

**Baseline Implementation Details.** For baselines, we include several inference-time augmentation methods that aim to improve visual information utilization without additional training. INST is implemented by replacing the base QA prompt with an augmented version that explicitly instructs the model to attend to visual evidence when answering questions:

> **Instructioning prompt template**
>
> Answer the question based on the image. Focus on the most relevant visual evidence in the image when generating your answer. If the image does not contain any relevant evidence, output "I cannot answer based on the given image." Image: {image} Question: {question}

CGR (Liu et al., 2025) follows a two-step reasoning process. In the first step, we prompt the model to produce a detailed caption or textual extraction of the image content using the following template:

> **CGR captioning prompt template**
>
> Carefully read and describe all relevant details from the image, especially any visible text or objects that may help answer the question. Provide a concise but detailed textual description of the image content. Image: {image}

The generated description is then concatenated with the original image and question, and the model generates the final answer using the base QA prompt. VAR (Liu et al., 2025) directly uses the final-layer attention scores to construct a binary mask that highlights salient image regions while suppressing less relevant ones. The original method applies the raw attention distribution without post-processing; in our implementation, we additionally apply a denoising and smoothing procedure (using the same parameters as in VEA) to improve stability. AGLA (An et al., 2025) is implemented using the official codebase, which is based on GradCAM (Selvaraju et al., 2017). It generates a saliency map, masks out irrelevant regions, and ensembles the outputs from the original image and the masked image. We adopt the default hyperparameter settings provided in the official implementation.

## B ADDITIONAL DISCUSSIONS

### B.1 USAGE OF ARTIFACTS AND AI ASSISTANTS

All models and datasets used in this study are publicly available on HuggingFace, and we adhered to their respective licenses and terms of use, limiting our work to non-commercial academic research. These models and datasets have been reviewed by their developers/creators to minimize the inclusion of personally identifiable information or offensive content and are widely adopted by the research community. We used AI tools to assist with language refinement during the writing process, the paper contains no AI-generated paragraphs. All material has been carefully reviewed to ensure accuracy and adherence to ethical standards.

### B.2 LIMITATIONS AND POTENTIAL SOLUTIONS.

While the proposed VEA framework demonstrates consistent improvements across diverse VLMs and tasks, several limitations remain. First, our method relies on extracting attention maps from Transformer layers, which requires access to intermediate activations during inference. This may not be supported by proprietary or API-only VLMs. Second, although our experiments provide strong empirical validation, attention may not always fully capture all the internal mechanisms of evidence localization, and alternative attribution signals (e.g., gradient-based saliency or probing features) could provide complementary insights. Third, while per-model profiling proves effective in identifying visual-grounding layers, more efficient or automated profiling strategies could further improve usability, especially for large-scale or real-time applications.

### B.3 FUTURE DIRECTIONS IN VLM (AGENTIC) REASONING AND BEYOND.

**Attention as Active Perception.** We believe that the internal attention signals of VLMs hold promising potential for broader applications in more complex multimodal reasoning scenarios. Beyond the image highlighting strategy explored in this work, attention signals could be leveraged to guide targeted image manipulations that facilitate reasoning in long-horizon or agentic tasks. For example, when a reasoning step requires the model to examine fine-grained details, attention could be used to **dynamically crop or zoom** into the relevant region of the image and feed it back into the context. Such adaptive image refinement guided by the model's own signals may allow VLMs to act more like human observers, selectively allocating focus as reasoning unfolds.

**Self-Triggered Visual Enhancement.** These mechanisms may be **combined with lightweight enhancement modules**, such as super-resolution or dehazing, to selectively refine local regions of interest with minimal computational cost. This opens up the possibility of multi-stage visual processing pipelines that are triggered only when the model itself recognizes uncertainty or insufficient evidence. Beyond single-image reasoning, attention-based guidance could also support multi-hop visual reasoning across multiple images or documents, where the model iteratively decides which visual regions to revisit or enhance.

**Toward Agentic Problem Solvers.** We view these directions as promising opportunities for enabling more efficient and adaptive agentic multimodal reasoning, where **models can actively exploit their own internal signals** to better support complex decision-making processes. In the longer term, embedding such self-directed evidence gathering into broader agentic frameworks could help VLMs evolve from passive perception systems into active problem solvers that can plan, verify, and refine their own reasoning steps.

## C ADDITIONAL RESULTS AND ANALYSIS

### C.1 ADDITIONAL EVALUATION ON MULTI-TURN AND MULTI-IMAGE REASONING

Beyond single-turn, single-image QA, real-world multimodal interactions often require reasoning across dialogue turns or integrating information from multiple images. These settings introduce additional challenges: the text context grows longer, grounding must remain consistent across turns, and evidence must be identified across multiple visual sources. To assess the generality of VEA under such conditions, we further evaluate it on representative multi-turn and multi-image benchmarks.

**Setup.** For multi-turn reasoning, we use the Visual Dialog (VisDial) dataset (Das et al., 2017), where each question needs to be answered using both the associated image and the cumulative dialogue history. We integrate the historical dialog in a new "dialog" section in the input prompt template. Following prior work, we report Token F1 and Exact Match (EM). VEA is applied using the attention from the first decoding step of the current turn. For multi-image reasoning, we evaluate the Semantic Correspondence task in the BLINK benchmark (Fu et al., 2024), which requires cross-image visual grounding beyond textual descriptions. Specifically, in this task, the first image gives the reference point, and the model is asked to identify the point from 4 possible options in the second image that is semantically matched to the reference point in the first image. We apply VEA on the second image and left the reference image untouched. Since the task is multiple-choice, we report Accuracy.

**Multi-Turn QA.** Table 5 presents the results on VisDial. VEA consistently improves both metrics across all tested models, with particularly notable gains in Token F1. VisDial answers are free-form, and many visually grounded improvements may not yield exact string matches. Thus, F1 provides a more sensitive measure of grounding quality. We observe that VEA shifts model predictions toward visually anchored content, increasing lexical overlap with ground truth even when EM remains unchanged. This indicates that VEA helps maintain consistent evidence usage across turns and reduces generic or image-agnostic responses.

Table 5: Multi-turn QA results on VisDial dataset.

| VisDial | LLaVA-1.6-7B | Qwen2.5VL-7B | Gemma3-4B | InternVL3.5-8B |
|---------|--------------|--------------|-----------|----------------|
| Base F1 | 28.5 | 27.5 | 25.0 | 33.2 |
| VEA F1 | 44.9 (+16.4) | 47.8 (+20.3) | 41.2 (+16.2) | 48.6 (+15.4) |
| Base EM | 38.4 | 37.6 | 34.0 | 38.5 |
| VEA EM | 40.9 (+2.5) | 40.5 (+2.9) | 37.5 (+3.5) | 41.4 (+2.9) |

**Multi-Image QA.** Table 6 summarizes the results on the BLINK Semantic Correspondence task. VEA again improves performance across all evaluated models. Multi-image reasoning requires identifying consistent semantic cues across distinct images. VEA enhances this process by suppressing irrelevant visual regions and strengthening attention to patches that share meaningful correspondences across images. Notably, even strong models such as InternVL3.5-8B benefit from VEA, demonstrating that inference-time evidence emphasis remains complementary to advanced multimodal architectures.

Table 6: Multi-image QA results on BLINK dataset.

| BLINK | LLaVA-1.6-7B | Qwen2.5VL-7B | Gemma3-4B | InternVL3.5-8B |
|-------|--------------|--------------|-----------|----------------|
| Base ACC | 34.4 | 61.4 | 42.2 | 66.4 |
| VEA ACC | 45.2 (+10.8) | 68.9 (+7.5) | 50.9 (+8.7) | 71.3 (+4.9) |

These additional results show that VEA generalizes robustly to more complex multimodal settings. Whether the model must track evidence across dialogue turns or align visual cues across multiple images, emphasizing visual evidence consistently improves grounding quality. This highlights the broad applicability of VEA as a lightweight inference-time enhancement for multimodal reasoning.

## C.2 EVALUATION ON GLOBAL-CONTEXT REASONING BENCHMARKS

As discussed in RQ4 (Fig. 7), overly aggressive masking (e.g., $\alpha = 1$) can remove too much background, but within a practical range ($\alpha \in [0.25, 0.75]$), VEA consistently enhances grounding without impairing global comprehension. We also include a variant, VEA$^*$, that preserves both the original and highlighted images to further strengthen global-context access. To further examine whether VEA affects performance in scenarios where the focus is on the non-text visual elements in global context, we additionally evaluate it on AI2D (Kembhavi et al., 2016) and MMStar (Chen et al., 2024). These datasets emphasize scientific or conceptual reasoning over fine-grained perception. However, we also note that their images typically consist of large, clean graphical elements where visual evidence localization is not the primary challenge.

**Experimental Setup.** AI2D focuses on diagram understanding and scientific concept reasoning, while MMStar aggregates multimodal scientific and mathematical problems, including samples from MMMU and MathVista. Since both benchmarks use multiple-choice questions, we report accuracy. For VEA*, the model receives both the original image and the VEA-highlighted image as input. Additionally, include a variant VEA* that retains both the original and highlighted images to check whether this can further strengthen access to global visual information.

Table 7: Results on global-context benchmarks (AI2D and MMStar).

| Dataset | Method | LLaVA-1.6-7B | Qwen2.5VL-7B | Gemma3-4B | InternVL3.5-8B |
|---------|--------|--------------|--------------|-----------|----------------|
| AI2D | Base | 66.4 | 78.9 | 56.2 | 77.2 |
| | VEA | 69.8 (+3.4) | 81.1 (+2.2) | 61.5 (+5.3) | 81.8 (+4.6) |
| | VEA* | 70.2 (+3.8) | 82.3 (+3.4) | 63.3 (+7.1) | 83.7 (+6.5) |
| MMStar | Base | 36.7 | 48.4 | 32.8 | 53.7 |
| | VEA | 40.5 (+3.8) | 52.8 (+4.4) | 37.5 (+4.7) | 57.6 (+3.9) |
| | VEA* | 40.2 (+3.5) | 53.3 (+4.9) | 39.1 (+6.3) | 58.2 (+4.5) |

**Results and Analysis.** Table 7 summarizes the results. The improvements on AI2D and MMStar confirm that VEA does not hinder global-context reasoning, though the magnitude of improvement is smaller compared with datasets that rely more heavily on fine-grained visual evidence. This aligns with the fact that these reasoning tasks emphasize logical inference rather than evidence localization. The VEA* variant further boosts performance by preserving full-image information while still highlighting evidence regions, particularly benefiting models with strong multi-image reasoning abilities (e.g., Qwen2.5-VL, InternVL). These results support that VEA is compatible with reasoning-centric benchmarks and that, within reasonable intervention strengths, foreground emphasis does not disrupt global context understanding.

## C.3 MULTI-ROUND SELF-VALIDATION WITH VEA

While VEA enhances visual grounding by highlighting evidence during a single inference pass, its decision process remains static, limited by the model's initial belief. In principle, a model could revisit its own outputs and revise incorrect responses, guided by the same visual attention mechanism in a cascaded fashion. Motivated by this, we explore whether equipping VEA with a self-validation loop, where the model reflects on and potentially corrects its previous answers, can further improve response accuracy.

**Experimental Setup.** We implement a variant, **VEA-cascade**, which adds a lightweight multi-round self-validation process. After generating an initial answer, the model receives the VEA-highlighted image and is asked to verify whether its own answer is correct. If the model responds affirmatively (e.g., "I confirm this answer is correct"), we take the answer as final. Otherwise, the model is prompted to revise the answer and regenerate the attention map for the next round. We allow up to 3 validation rounds and conduct experiments using Qwen2.5-VL-7B on four datasets.

Table 8: Performance of VEA-cascade with multi-round self-validation.

| Method | TextVQA | DocVQA | SROIE | InfographicsQA |
|--------|---------|--------|-------|----------------|
| VEA EM | 90.3 | 76.2 | 82.5 | 64.4 |
| VEA-cascade EM | 90.3 (+0.0) | 76.6 (+0.4) | 82.7 (+0.2) | 64.4 (+0.0) |
| Avg. #Rounds | 1.05 | 1.09 | 1.10 | 1.07 |

**Results and Analysis.** Table 8 summarizes the results. While VEA-cascade introduces self-reflection, it leads to only marginal gains across all tasks, and the average number of rounds remains close to 1. This suggests the model rarely alters its initial response, even when given the opportunity to revise. The limited improvements from VEA-cascade reveal that current VLMs tend to reaffirm their initial answers, with only minor evidence of genuine self-correction. The small number of rounds used suggests the model lacks confidence calibration or uncertainty estimation necessary for productive self-reflection. These findings align with recent studies that highlight the challenge of inducing

effective reflection in VLMs Wei et al. (2025); Jiang et al. (2025). Nonetheless, we believe that by integrating VEA's attention feedback with stronger multi-step reasoning modules and more structured reflection prompts, future work could unlock richer and more adaptive visual reasoning pipelines, as discussed in Section B.3.

### C.4 WHEN ATTENTION IS NOT RELIABLE.

VEA relies on the model's internal attention maps to locate supporting visual evidence. While prior analyses confirm that VLM attention often aligns well with ground-truth evidence, it remains important to examine failure modes, particularly when attention grounding is inaccurate. We aim to answer three questions: (1) How frequently does attention misalign with ground truth? (2) Can VEA still improve performance in such cases? (3) What are the underlying causes of grounding failures?

Table 9: VEA performance on samples with low attention-evidence alignment (AUROC < 0.5).

| Metric | TextVQA | DocVQA | SROIE | InfographicsQA |
|---|---|---|---|---|
| Low-AUROC Ratio (%) | 1.42 | 3.11 | 7.34 | 5.20 |
| Base F1 | 15.3 | 16.2 | 10.6 | 27.6 |
| VEA F1 | 21.8 (+6.5) | 26.4 (+10.2) | 18.9 (+8.3) | 35.3 (+7.7) |

**Results and Discussions.** We identify "low-AUROC" samples where attention does not correlate well with labeled evidence (AUROC < 0.5) with Qwen2.5VL-7B. Table 9 shows that such cases are rare, ranging from 1.4% to 7.3% across datasets, and that VEA still improves QA performance even on these samples. Despite poor AUROC, VEA often leads to better answers. Upon manual inspection, we find two main causes for these "false failure" cases: **(i) Incomplete annotations.** For many questions, the image contains multiple valid evidence regions, but only one is labeled in the ground truth. When the model correctly attends to unlabeled regions, AUROC is low despite accurate grounding. VEA remains beneficial in such scenarios, as it reinforces these valid regions. **(ii) Genuine grounding failure.** In a small number of cases, the model fails to perceive any correct visual evidence and highlights irrelevant regions. These typically stem from vague or misaligned annotations. For example, questions referencing nonexistent elements or ambiguous phrasing. In such cases, both the model's attention and VEA's highlights reflect dataset limitations rather than model deficiencies.

### C.5 DISCUSSION ON DISCREPANCIES WITH EXISTING FINDINGS

Several recent studies have reported seemingly different attention distribution trends. Upon closer examination, we find that these discrepancies arise mainly from differences in scope and terminology rather than fundamental disagreement. Our results are in fact consistent with the attention dynamics described in earlier work once these distinctions are accounted for.

Specifically, Amara et al. Amara et al. (2024) do not perform a layer-wise breakdown but instead measure the total attention mass allocated to image tokens, concluding that image tokens overall receive more attention than text tokens. This matches our findings, as shown in our layer-averaged analysis. Lu et al. Lu et al. (2025) report that early layers show a high visual-attention ratio with low concentration (broad scanning), and that middle layers maintain a high ratio while increasing concentration as cross-modal alignment emerges. This pattern directly corresponds to our observations: our "early" layers also exhibit diffuse visual scanning, and our "deeper" layers (roughly 50%–75% depth) align with their "middle" stages where focused visual grounding occurs. Similar layer trends are also reported by Bi et al. Bi et al. (2025), supporting that our interpretation is consistent with the broader literature.

Overall, these analyses reflect complementary perspectives rather than conflicting findings. Once methodological framing is harmonized, the attention evolution patterns described across studies present a coherent view of how multimodal transformers progressively shift from broad visual exploration to localized, semantically grounded alignment.

Table 10: Mean evidence attribution AUROC under different layer selection strategies.

| Layers / Method | LLaVA-7B | LLaVA-13B | Qwen-7B | Qwen-32B | Gemma-4B | Gemma-27B |
|---|---|---|---|---|---|---|
| $L_{0\%-100\%}$ | 75.9 | 76.3 | 68.5 | 57.0 | 59.5 | 61.8 |
| $L_{0\%-50\%}$ | 68.2 | 73.1 | 59.4 | 51.3 | 56.5 | 55.4 |
| $L_{50\%-100\%}$ | 78.0 | 76.9 | 79.5 | 67.6 | 65.9 | 68.1 |
| $L_{50\%-75\%}$ | **81.2** | **79.1** | **80.3** | 65.9 | **71.7** | **70.0** |
| $L_{75\%-100\%}$ | 74.8 | 74.7 | 78.7 | **70.4** | 60.8 | 66.3 |
| VEA Profiling | **83.6** | **84.4** | **85.2** | **79.1** | **80.0** | **81.2** |

Table 11: TextVQA Token F1 under different layer selection strategies.

| Layers / Method | LLaVA-7B | LLaVA-13B | Qwen-7B | Qwen-32B | Gemma-4B | Gemma-27B |
|---|---|---|---|---|---|---|
| Base | 27.8 | 66.8 | 79.5 | 72.1 | 58.2 | 57.6 |
| $L_{0\%-100\%}$ | 58.2 | 72.8 | 86.6 | 78.2 | 64.5 | 61.2 |
| $L_{0\%-50\%}$ | 38.9 | 71.6 | 80.1 | 71.8 | 59.9 | 56.4 |
| $L_{50\%-100\%}$ | 62.2 | 73.1 | **87.6** | 79.8 | 65.4 | 62.0 |
| $L_{50\%-75\%}$ | **66.7** | **74.1** | 87.5 | 78.8 | **66.9** | **62.1** |
| $L_{75\%-100\%}$ | 56.5 | 71.9 | 87.1 | **80.6** | 63.6 | **62.1** |
| VEA Profiling | **69.4** | **76.3** | **88.6** | **81.4** | **69.6** | **63.2** |

## C.6 EVALUATION ON THE EFFECT OF EVIDENCE-LAYER SELECTION

VEA highlights image regions by manipulating attention at specific transformer layers. Since different VLMs may exhibit grounding behavior at different depths, we identify high-evidence layers for each model. This section discusses (1) what is the benefit of performing profiling for each model, and (2) how sensitive VEA is to inaccurate or static layer choices.

**Profiling helps achieve more accurate evidence identification.** Table 10 reports the average evidence attribution AUROC across four datasets (following the setup in Table 2) under different layer selection strategies. In addition to the 3 static layer choices in Table 2, we further include two finer-grained static ranges, $L_{50\%-75\%}$ and $L_{75\%-100\%}$. Across six model families, the later half of the layers ($L_{50\%-100\%}$) generally produces stronger attribution than early layers, with $L_{50\%-75\%}$ often being the best-performing static choice. Per-model profiling further improves robustness and consistently achieves the highest AUROC, especially for models such as Gemma that exhibit non-monotonic, "periodic" grounding patterns across layer depth.

**Robustness to Inaccurate Selection.** We further examine VEA's sensitivity to imperfect layer identification by testing QA performance on TextVQA with intentionally static (non-profiled) layer choices. As shown in Table 11, VEA maintains strong gains even when the chosen layer range deviates from the optimal one. Using middle-to-deep layers ($L_{50\%-100\%}$) consistently yields stable improvements, while profiling provides additional but incremental benefits. Only early-layer steering ($L_{0\%-50\%}$) noticeably reduces performance, confirming that early layers lack visual grounding.

These results show that model-specific evidence-layer profiling is not a strict requirement but an enhancement. VEA remains effective even with coarse, static layer choices, confirming its robustness to layer selection inaccuracies. Profiling, however, helps align VEA's steering with model-specific grounding depths and mitigates variance across architectures, particularly for models with irregular attention-layer behavior.

## C.7 MORE VISUAL EXAMPLES

Figure 8 provides additional qualitative examples with Qwen2.5VL-7B to illustrate VEA's behavior across diverse visual question answering scenarios. These results show that VEA consistently enhances visual grounding and answer accuracy by guiding models to focus on semantically relevant regions, regardless of scene type or task focus.

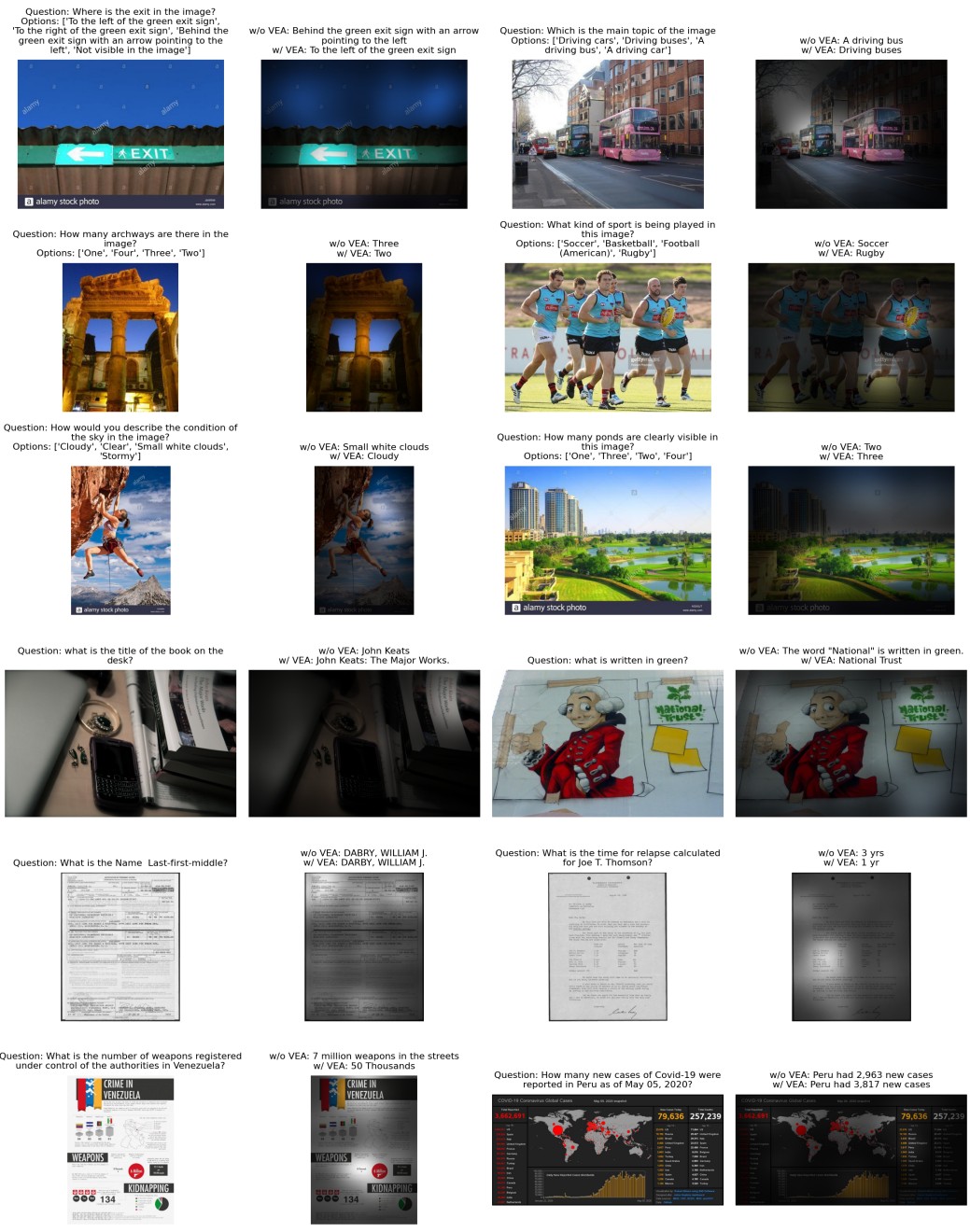

Figure 8: Additional qualitative examples of VEA. For diversity, the upper half of the examples are drawn from the MMStar dataset, which focuses on natural scenes, while the lower half are taken from the four datasets used in the main experiments, which emphasize textual or document-centric scenes.

## C.8    FULL QUESTION ANSWERING RESULTS

Table 12 reports the complete results of all methods across four VQA datasets and eight VLMs. We observe that VEA consistently achieves the best or second-best performance in nearly all settings, often ranking first in both *Exact Match (EM)* and *Token F1*. Compared with the base models, VEA delivers substantial gains, particularly on datasets that require fine-grained reading comprehension such as TextVQA and DocVQA. Among the baselines, AGLA and CGR occasionally achieve competitive performance, but their improvements are less stable across model families and scales. In contrast,

Table 12: Full results of applying visual evidence augmentation methods to 8 VLMs from 4 latest series of LLaVA, Qwen, Gemma, and InternVL families. We report detailed results on four visual question answering tasks, including both *Exact Match (EM)* and *Token F1* scores. For each metric, we also report the average rank of each method over all tested tasks and models. These results complement the Table 1 in the paper and provide a comprehensive view of performance across datasets and metrics.

| Model | | Method | TextVQA | | DocVQA | | SROIE | | InfographicsQA | | Average Rank (↓) | |
|---|---|---|---|---|---|---|---|---|---|---|---|---|
| | | | EM | Token F1 | EM | Token F1 | EM | Token F1 | EM | Token F1 | EM | Token F1 |
| LLaVaNext | 7B | BASE | 48.44 | 27.78 | 44.53 | 46.74 | 41.41 | 46.17 | 19.53 | 12.51 | 5.50 | 5.50 |
| | | INST | 47.84 | 43.95 | 46.71 | 46.49 | 41.00 | 45.35 | 19.82 | 17.98 | 5.25 | 5.25 |
| | | CGR | 66.50 | 41.03 | 45.89 | 51.21 | 42.85 | 52.39 | 26.54 | 19.52 | 3.00 | 3.50 |
| | | VAR | 55.16 | 63.65 | 48.61 | 48.15 | 42.75 | 51.56 | 24.10 | 19.51 | 3.50 | 3.50 |
| | | AGLA | 65.39 | 49.96 | 48.78 | 51.41 | 42.24 | 53.79 | 29.07 | 25.31 | 2.75 | 2.00 |
| | | VEA | 75.32 | 69.36 | 49.24 | 53.51 | 43.89 | 53.78 | 29.86 | 25.60 | 1.00 | 1.25 |
| | 13B | BASE | 69.53 | 66.76 | 56.25 | 64.43 | 43.75 | 59.86 | 28.12 | 24.05 | 5.25 | 5.75 |
| | | INST | 74.08 | 68.30 | 55.30 | 64.55 | 43.38 | 61.54 | 27.86 | 23.39 | 5.75 | 4.75 |
| | | CGR | 77.39 | 73.85 | 56.45 | 64.44 | 45.76 | 61.11 | 31.39 | 27.61 | 3.00 | 4.00 |
| | | VAR | 74.99 | 69.05 | 57.21 | 65.31 | 44.40 | 62.48 | 29.35 | 26.07 | 3.25 | 3.50 |
| | | AGLA | 78.06 | 74.40 | 57.12 | 66.17 | 44.44 | 62.61 | 34.01 | 31.74 | 2.25 | 1.75 |
| | | VEA | 78.40 | 76.25 | 57.06 | 66.21 | 46.34 | 62.65 | 34.50 | 31.63 | 1.50 | 1.25 |
| Qwen2.5 | 7B | BASE | 85.94 | 79.49 | 73.44 | 79.71 | 75.78 | 92.53 | 58.59 | 59.03 | 5.50 | 5.25 |
| | | INST | 85.31 | 80.26 | 75.06 | 80.02 | 75.63 | 91.24 | 59.50 | 58.88 | 5.25 | 5.25 |
| | | CGR | 88.69 | 86.42 | 73.59 | 79.97 | 80.98 | 93.66 | 60.98 | 63.39 | 3.75 | 3.50 |
| | | VAR | 87.82 | 81.56 | 75.70 | 80.81 | 78.79 | 91.64 | 63.42 | 61.83 | 3.25 | 4.00 |
| | | AGLA | 89.75 | 88.32 | 76.14 | 81.55 | 82.47 | 94.32 | 63.10 | 63.51 | 2.25 | 1.75 |
| | | VEA | 90.33 | 88.63 | 76.24 | 81.47 | 82.51 | 94.38 | 64.35 | 63.90 | 1.00 | 1.25 |
| | 32B | BASE | 81.25 | 72.10 | 73.44 | 74.65 | 68.75 | 81.43 | 53.91 | 51.12 | 5.50 | 6.00 |
| | | INST | 80.58 | 75.23 | 73.48 | 74.79 | 67.95 | 81.77 | 54.18 | 52.97 | 5.50 | 5.00 |
| | | CGR | 84.44 | 76.66 | 76.52 | 78.95 | 70.89 | 90.17 | 60.19 | 53.55 | 2.75 | 3.25 |
| | | VAR | 81.64 | 77.05 | 74.50 | 76.42 | 70.77 | 88.36 | 54.71 | 53.88 | 4.00 | 3.50 |
| | | AGLA | 84.75 | 77.30 | 75.06 | 80.00 | 74.79 | 89.10 | 60.47 | 54.26 | 2.25 | 2.00 |
| | | VEA | 86.87 | 81.44 | 77.03 | 80.45 | 76.02 | 91.47 | 63.32 | 54.23 | 1.00 | 1.25 |
| Gemma3 | 4B | BASE | 78.12 | 58.21 | 58.59 | 54.34 | 54.69 | 66.13 | 35.16 | 23.47 | 5.00 | 5.50 |
| | | INST | 76.93 | 61.76 | 59.06 | 57.90 | 53.90 | 64.88 | 34.29 | 23.09 | 5.75 | 5.50 |
| | | CGR | 82.53 | 63.72 | 63.65 | 61.50 | 57.78 | 67.45 | 37.55 | 26.27 | 2.50 | 3.25 |
| | | VAR | 81.57 | 68.29 | 60.36 | 60.41 | 54.96 | 69.83 | 34.95 | 27.09 | 4.25 | 2.50 |
| | | AGLA | 83.02 | 67.48 | 64.25 | 60.07 | 57.20 | 70.47 | 37.26 | 25.55 | 2.50 | 3.25 |
| | | VEA | 83.18 | 69.60 | 64.52 | 63.24 | 58.76 | 71.43 | 38.32 | 27.12 | 1.00 | 1.00 |
| | 27B | BASE | 85.16 | 57.58 | 70.31 | 67.26 | 70.31 | 82.88 | 51.56 | 22.88 | 6.00 | 5.50 |
| | | INST | 86.24 | 57.49 | 71.80 | 66.68 | 70.67 | 83.72 | 51.98 | 25.17 | 5.00 | 5.50 |
| | | CGR | 87.08 | 62.10 | 72.69 | 68.33 | 73.45 | 85.51 | 55.88 | 29.38 | 3.75 | 3.25 |
| | | VAR | 88.86 | 63.18 | 73.26 | 69.12 | 72.47 | 86.54 | 59.06 | 26.25 | 2.50 | 2.50 |
| | | AGLA | 91.48 | 63.15 | 72.85 | 67.82 | 74.16 | 86.54 | 58.43 | 28.20 | 2.50 | 3.00 |
| | | VEA | 91.84 | 63.22 | 75.23 | 69.11 | 75.19 | 86.87 | 59.00 | 29.95 | 1.25 | 1.25 |
| InternVL3.5 | 8B | BASE | 83.59 | 79.41 | 85.94 | 88.50 | 84.38 | 93.40 | 63.28 | 57.20 | 5.00 | 5.50 |
| | | INST | 83.30 | 80.31 | 86.43 | 89.56 | 83.17 | 92.18 | 63.87 | 57.95 | 5.50 | 5.25 |
| | | CGR | 86.58 | 84.34 | 87.64 | 91.47 | 84.12 | 92.74 | 71.16 | 69.48 | 3.00 | 3.75 |
| | | VAR | 83.67 | 82.80 | 87.08 | 89.89 | 84.53 | 94.37 | 65.33 | 69.87 | 3.50 | 3.00 |
| | | AGLA | 86.21 | 84.78 | 87.44 | 92.47 | 84.30 | 93.69 | 71.27 | 69.95 | 3.00 | 2.25 |
| | | VEA | 87.57 | 86.47 | 88.42 | 92.57 | 84.66 | 94.23 | 72.12 | 70.07 | 1.00 | 1.25 |
| | 14B | BASE | 86.72 | 75.06 | 88.28 | 82.31 | 80.47 | 79.48 | 61.72 | 40.94 | 5.25 | 5.25 |
| | | INST | 86.37 | 75.29 | 87.31 | 81.39 | 79.50 | 78.51 | 62.12 | 40.86 | 5.75 | 5.75 |
| | | CGR | 89.64 | 76.25 | 89.35 | 83.44 | 82.62 | 80.88 | 65.08 | 41.50 | 3.00 | 3.00 |
| | | VAR | 87.48 | 77.87 | 90.42 | 82.36 | 80.71 | 79.50 | 62.95 | 41.60 | 3.25 | 3.25 |
| | | AGLA | 90.04 | 76.15 | 90.23 | 83.11 | 84.96 | 81.68 | 63.86 | 42.33 | 2.50 | 2.50 |
| | | VEA | 90.21 | 78.15 | 90.24 | 84.17 | 85.35 | 81.67 | 65.73 | 42.94 | 1.25 | 1.25 |

VAR benefits from leveraging attention scores but remains sensitive to noisy signals, even with additional smoothing. Another key observation is that VEA provides consistent improvements across all four VLM families (LLaVA, Qwen, Gemma, and InternVL). The average rank results confirm this trend: VEA outperforms all other methods with the lowest ranks across metrics, demonstrating its robustness and generality as an inference-time augmentation strategy.

Table 13: Full results of visual evidence attribution accuracy across four VQA datasets. We report token-level *AUROC* and *NDCG@all* for different attribution methods, including both static layer-based baselines (e.g., $L_{0\%-100\%}$, $L_{0\%-50\%}$, $L_{50\%-100\%}$) and adaptive attribution approaches (VAR, AGLA, and VEA). For each dataset and model, we present results separately, and also include the average rank of each method across models. These results complement Table 2 in the main paper.

| | Evidence Attribution | LLaVA 7B | | LLaVA 13B | | Qwen 7B | | Qwen 32B | | Gemma 4B | | Gemma 27B | | Avg. Rank (↓) | |
|---|---|---|---|---|---|---|---|---|---|---|---|---|---|---|---|
| | | AUROC | NDCG | AUROC | NDCG | AUROC | NDCG | AUROC | NDCG | AUROC | NDCG | AUROC | NDCG | AUROC | NDCG |
| **TextVQA** | $L_{0\%-100\%}$ | 87.2 | 51.4 | 87.0 | 52.1 | 83.4 | 51.5 | 73.6 | 33.7 | 65.1 | 29.5 | 70.1 | 30.3 | 4.50 | 4.50 |
| | $L_{0\%-50\%}$ | 79.9 | 45.0 | 84.6 | 51.1 | 73.4 | 34.6 | 65.1 | 31.3 | 62.6 | 28.1 | 61.4 | 28.5 | 5.67 | 5.67 |
| | $L_{50\%-100\%}$ | 88.0 | 64.1 | 87.3 | 56.6 | 86.7 | 63.9 | 81.1 | 47.5 | 68.2 | 35.8 | 76.6 | 38.8 | 2.83 | 2.83 |
| | VAR | 82.0 | 49.5 | 84.2 | 46.3 | 84.1 | 61.3 | 80.6 | 43.1 | 54.5 | 26.8 | 71.8 | 33.0 | 4.83 | 4.67 |
| | AGLA | 90.2 | 68.4 | 90.4 | 66.8 | 86.2 | 61.0 | 87.0 | 58.0 | 72.4 | 36.8 | 81.9 | 43.9 | 2.17 | 2.33 |
| | **VEA** | **92.2** | **76.5** | **92.4** | **75.3** | **90.7** | **74.2** | **89.8** | **63.9** | **88.0** | **63.2** | **88.0** | **55.0** | **1.00** | **1.00** |
| **DocVQA** | $L_{0\%-100\%}$ | 76.6 | 45.2 | 77.7 | 46.3 | 67.2 | 40.9 | 53.2 | 33.2 | 60.2 | 37.8 | 62.0 | 38.8 | 4.33 | 4.33 |
| | $L_{0\%-50\%}$ | 66.5 | 41.5 | 72.5 | 44.8 | 59.8 | 35.5 | 50.5 | 32.8 | 57.2 | 36.5 | 56.4 | 37.2 | 5.67 | 5.67 |
| | $L_{50\%-100\%}$ | 79.6 | 53.7 | 79.2 | 50.2 | 80.0 | 59.7 | 66.1 | 45.0 | 67.6 | 48.2 | 67.7 | 46.6 | 2.83 | 2.83 |
| | VAR | 70.3 | 43.2 | 74.5 | 43.8 | 75.8 | 54.9 | 65.8 | 41.7 | 52.0 | 34.7 | 55.6 | 45.9 | 5.00 | 5.00 |
| | AGLA | 81.7 | 56.3 | 83.0 | 56.0 | 77.8 | 56.8 | 73.8 | 54.0 | 69.5 | 49.2 | 73.7 | 52.7 | 2.17 | 2.17 |
| | **VEA** | **85.2** | **63.1** | **85.7** | **62.9** | **86.0** | **70.4** | **76.5** | **60.8** | **78.7** | **61.8** | **81.9** | **64.8** | **1.00** | **1.00** |
| **SROIE** | $L_{0\%-100\%}$ | 71.7 | 58.6 | 70.0 | 56.5 | 62.3 | 44.1 | 51.5 | 38.8 | 58.8 | 44.5 | 60.7 | 45.5 | 4.33 | 4.33 |
| | $L_{0\%-50\%}$ | 63.8 | 54.5 | 67.6 | 56.2 | 53.9 | 40.4 | 44.4 | 37.9 | 54.5 | 43.0 | 53.0 | 43.7 | 5.67 | 5.67 |
| | $L_{50\%-100\%}$ | 72.7 | 63.2 | 70.7 | 59.6 | 73.4 | 59.3 | 62.2 | 46.7 | 65.6 | 54.5 | 64.7 | 52.4 | 2.83 | 2.83 |
| | VAR | 65.8 | 54.9 | 63.9 | 53.3 | 65.5 | 53.5 | 56.8 | 43.9 | 49.9 | 42.1 | 54.0 | 44.9 | 5.00 | 5.00 |
| | AGLA | 75.0 | 65.7 | 75.6 | 65.0 | 71.7 | 57.1 | 68.9 | 53.0 | 67.8 | 55.4 | 70.9 | 58.1 | 2.17 | 2.17 |
| | **VEA** | **78.9** | **72.6** | **79.0** | **71.1** | **80.4** | **68.9** | **74.2** | **59.7** | **80.2** | **69.8** | **78.5** | **68.5** | **1.00** | **1.00** |
| **InfographVQA** | $L_{0\%-100\%}$ | 68.3 | 33.6 | 70.3 | 34.1 | 61.2 | 30.3 | 49.8 | 26.4 | 53.8 | 30.3 | 54.4 | 30.9 | 4.17 | 4.50 |
| | $L_{0\%-50\%}$ | 62.4 | 31.7 | 67.5 | 33.4 | 50.6 | 26.3 | 45.4 | 25.7 | 52.0 | 29.7 | 50.9 | 30.0 | 5.67 | 5.67 |
| | $L_{50\%-100\%}$ | 71.5 | 37.2 | 70.3 | 35.5 | 78.0 | 49.6 | 61.1 | 33.8 | 62.3 | 36.2 | 63.4 | 36.6 | 3.00 | 2.83 |
| | VAR | 65.3 | 32.9 | 65.8 | 32.6 | 75.6 | 46.8 | 59.7 | 30.4 | 48.5 | 29.5 | 51.5 | 31.5 | 4.83 | 4.83 |
| | AGLA | 73.8 | 38.4 | 75.5 | 38.5 | 75.1 | 46.8 | 70.6 | 40.3 | 63.6 | 36.6 | 68.9 | 41.2 | 2.33 | 2.17 |
| | **VEA** | **77.9** | **42.0** | **80.6** | **44.5** | **83.9** | **60.9** | **75.7** | **49.2** | **73.0** | **45.0** | **76.4** | **52.0** | **1.00** | **1.00** |

## C.9 FULL EVIDENCE ATTRIBUTION ACCURACY RESULTS

Table 13 reports the full token-level attribution results across four datasets and six representative VLMs. Several consistent patterns can be observed. First, static baselines that aggregate attention uniformly over either all layers ($L_{0\%-100\%}$) or early layers ($L_{0\%-50\%}$) perform poorly, indicating that not all layers contribute equally to evidence attribution. In contrast, restricting attention to later layers ($L_{50\%-100\%}$) yields substantially better AUROC and NDCG scores, confirming that deeper layers play a critical role in localizing evidence. Second, adaptive attribution methods such as VAR and AGLA improve upon static baselines in many cases, with AGLA in particular showing stronger ranking quality across datasets. Finally, VEA with layer profiling consistently achieves the highest scores in both AUROC and NDCG@all across all models and tasks, leading to the best overall ranking. These results validate that VEA more accurately identifies relevant evidence tokens compared with both static and adaptive alternatives, and demonstrate the robustness of the proposed approach across diverse datasets and model families.

## C.10 FULL LAYER-WISE ATTENTION DYNAMICS VISUALIZATION

We also provide the completion of the layer-wise attention dynamics visualizations in Section 2.

Figure 9, as an extension of Figure 1 in Section 2.1, presents the layer-wise modality attention transition across multiple models and datasets. Although the detailed patterns differ across model families, the overall trend holds consistently: early layers predominantly attend to text tokens, whereas deeper layers progressively shift their focus toward image tokens, revealing a sequential transition from linguistic parsing to visual grounding within single-token inference.

Figure 10, complementing Figure 4 in Section 2.3, shows the relative average attention assigned to evidence versus non-evidence image tokens across layers. Consistent with our main-text analysis, deeper layers across all models and datasets consistently allocate higher attention to evidence regions, even in cases where the model produces incorrect answers.

To more directly assess the ability of different layers to locate visual evidence, Figure 11 reports the evidence attribution accuracy (*AUROC* and *NDCG@all*) of each layer across multiple models and datasets. In line with Figure 10 and prior discussions, deeper layers generally achieve higher attribution accuracy. Interestingly, however, the distribution of optimal layers varies across model families: LLaVA's best-performing layers cluster around the middle layers, Qwen's peak layers concentrate near the final output layers, while Gemma exhibits a periodic pattern in which every few layers contain a "good attribution layer." These diverse patterns highlight the importance and benefit of per-model profiling. We hypothesize that such differences may stem from family-specific design choices or training strategies, though a deeper understanding of their underlying causes remains open for future investigation.

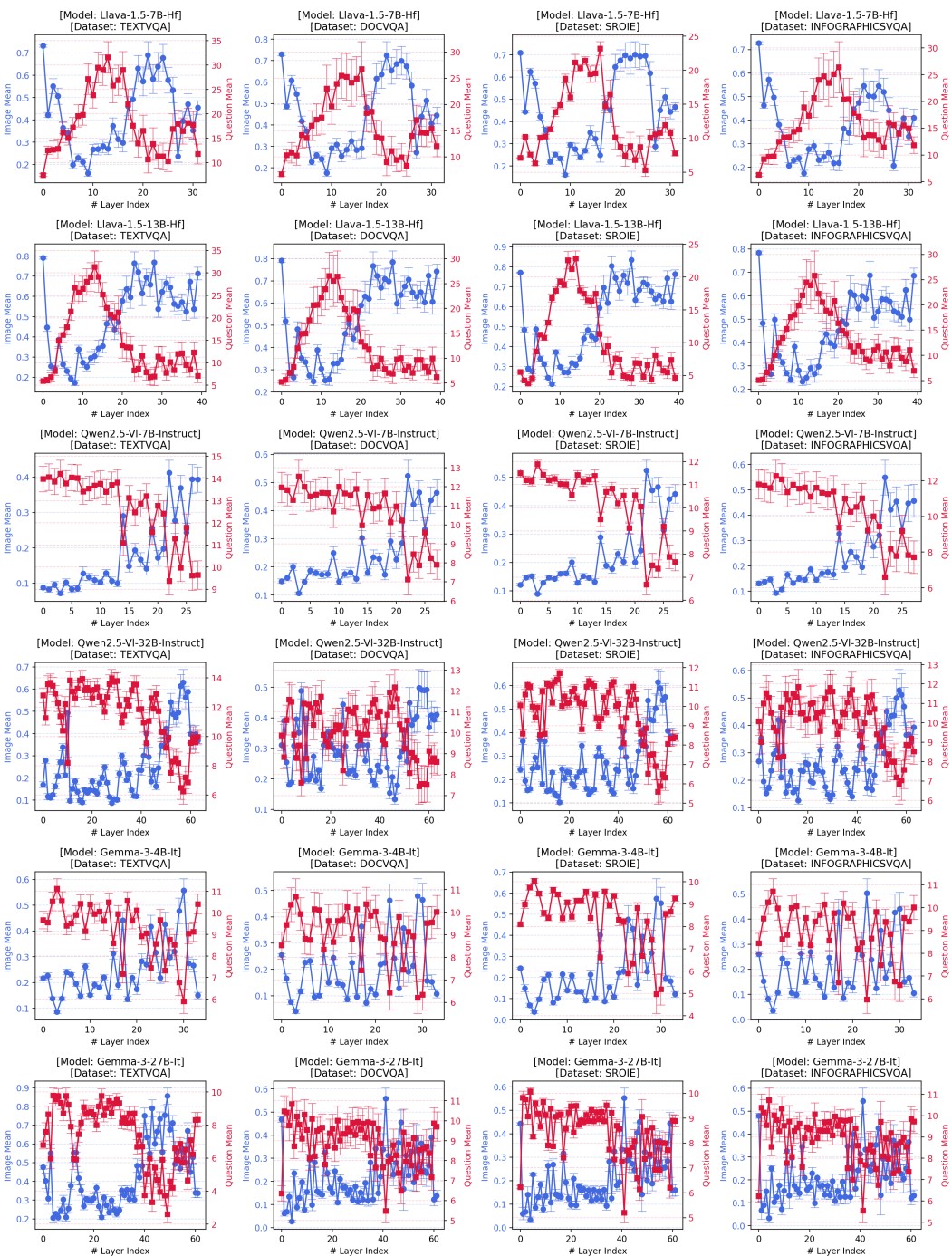

Figure 9: Relative attention per token (RAPT) (y-axis) to text tokens (red) and image tokens (blue) across Transformer layers for six representative VLMs (LLaVA-7B, LLaVA-13B, Qwen2.5-VL-7B, Qwen2.5-VL-32B, Gemma3-4B, Gemma3-27B) on four VQA datasets. Across all models and datasets, we observe a consistent trend: early layers attend predominantly to text tokens, whereas deeper layers gradually increase their focus on image tokens. This reveals a sequential transition from linguistic parsing to visual grounding during single-token inference. These figures complement Figure 1 in the main paper.

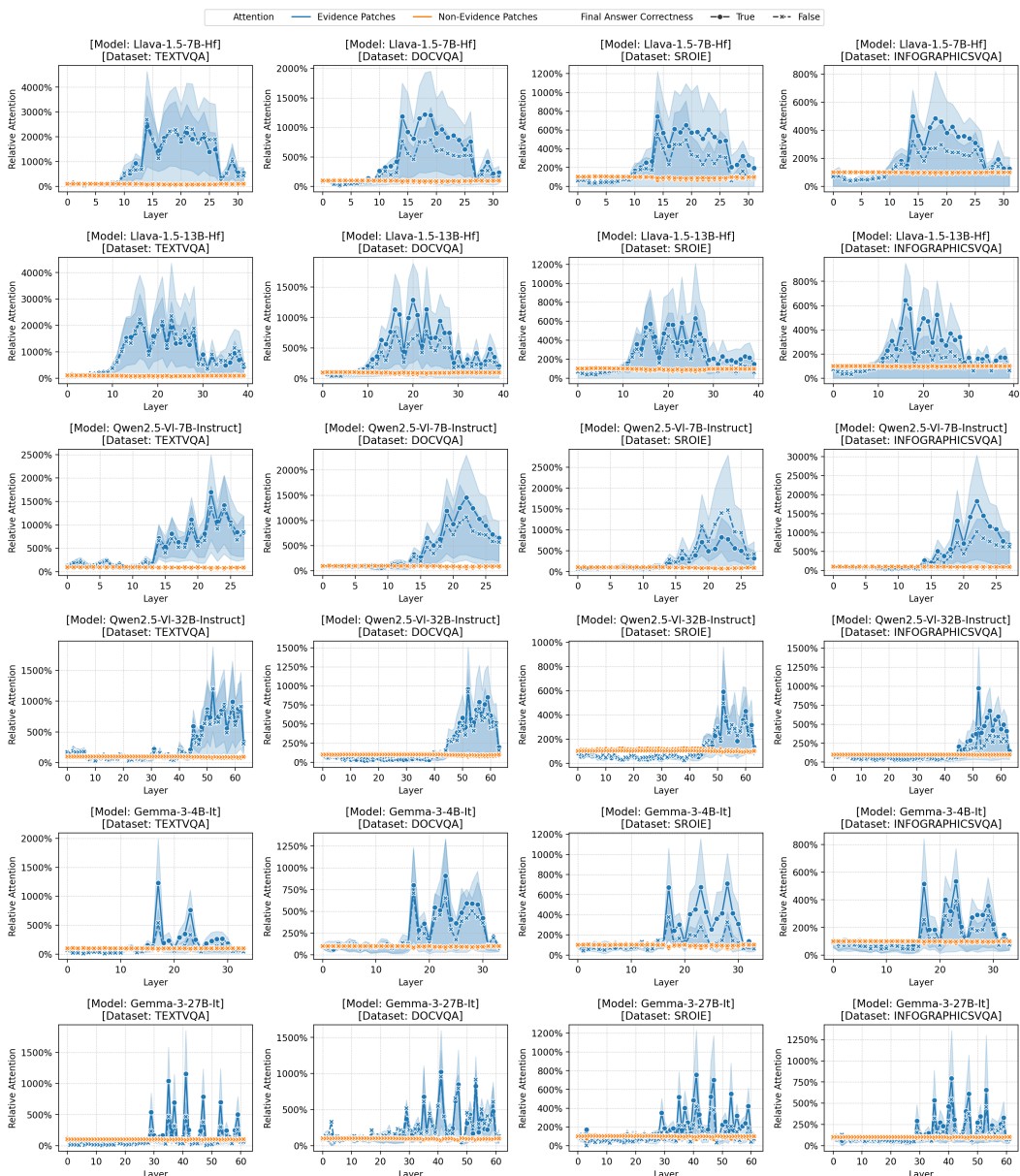

Figure 10: Relative attention to evidence image tokens (blue) vs. non-evidence image tokens (orange) across layers for six VLMs (LLaVA-7B/13B, Qwen2.5-VL-7B/32B, Gemma3-4B/27B) on four VQA datasets. Across models and datasets, deeper layers consistently assign higher attention to evidence regions, even when answers are incorrect (dashed lines). Best viewed in color. These figures complement Figure 4 in the main paper.

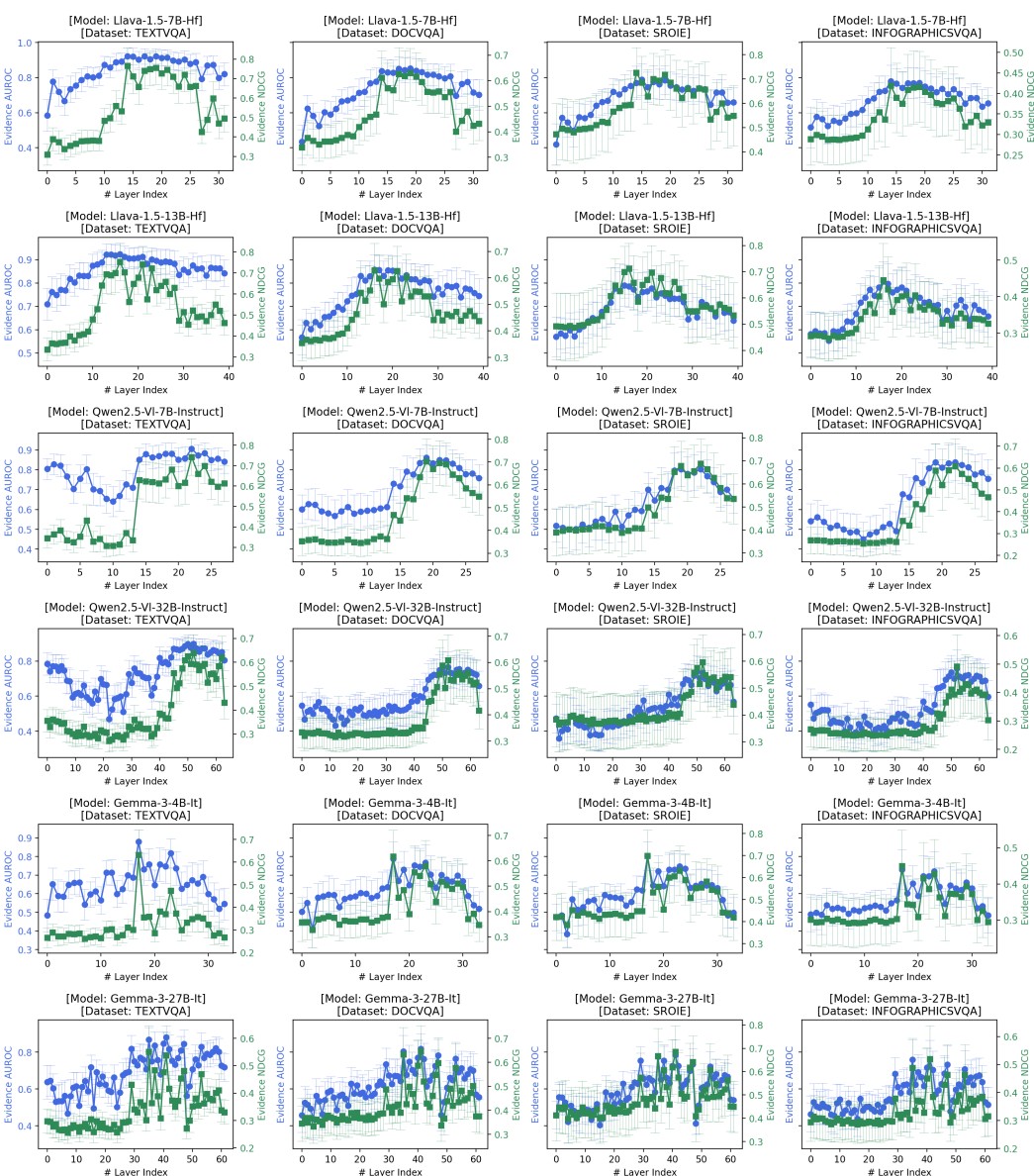

Figure 11: Layer-wise evidence attribution accuracy (AUROC and NDCG) of six VLMs (LLaVA-7B/13B, Qwen2.5-VL-7B/32B, Gemma3-4B/27B) on four VQA datasets. Consistent with Figure 10, deeper layers generally achieve higher attribution accuracy. While the optimal layers vary across models, their patterns are stable across datasets, highlighting the benefit of per-model profiling.

