# OpenReview forum: "Seeing but Not Believing: Probing the Disconnect Between Visual Attention and Answer Correctness in VLMs"
_ICLR.cc/2026/Conference — ICLR 2026 Poster_

### Official Review · Reviewer_3b67 · 2025-10-22

**Soundness:** 3
**Presentation:** 3
**Contribution:** 2
**Rating:** 2
**Confidence:** 4

**Summary:**

The paper investigates the cause of hallucinations in Vision-Language Models, hypothesizing that these arise because models perceive visual information but fail to leverage it effectively. The authors analyze attention distributions across layers, observing that early layers predominantly attend to textual features, while deeper layers focus more on visual evidence, and this focus is not always used for reasoning. To address this, they propose a steering mechanism that explicitly increases attention to visual evidence regions, enhancing the model’s grounding on visual cues.

**Strengths:**

- The paper offers an insightful analysis of attention localization across layers, showing that deeper layers indeed attend more to semantically relevant image regions when forming answers.

- The VEA step is methodologically well-designed, particularly with its denoising and Gaussian smoothing components to maintain spatial coherence in the attention masks. The method provides a simple yet interpretable tool for reducing hallucinations.

**Weaknesses:**

- The comparison between text and image RATP values (Fig. 1) is not clearly justified, given that the two modalities operate on different attention scales. It remains unclear how a small increase in image RATP can be interpreted as a “modality shift,” especially when text attention is still orders of magnitude higher (e.g., 0.2 → 0.6 for image vs. ~20 for text). This weakens the claim that “vision plays a stronger role in later inference stages.”

- Prior studies (e.g., [1], [2]) have reported different attention trends, showing that image tokens already dominate attention in early layers. The authors should better situate their findings within this literature, clarifying methodological or model-related differences that explain these discrepancies.

- The interpretation of Fig. 4 could be refined: while the authors claim that evidence tokens receive higher attention even in incorrect answers, the plot also shows a drop in attention between correct and incorrect predictions, which complicates that conclusion.

- The assumption that deep layers consistently capture “ground-truth” evidence regions is empirically plausible but not guaranteed. The paper would benefit from a clearer discussion on the reliability of these “evidence layers” across architectures and datasets.

- It is also unclear whether the identification of such evidence layers must be repeated per model, and how sensitive the steering mechanism is to inaccuracies in this identification.

[1] Amara, Kenza, et al. "Why context matters in VQA and Reasoning: Semantic interventions for VLM input modalities." arXiv preprint arXiv:2410.01690 (2024).

[2] Lu, Haolang, et al. "Mitigating Hallucination in Multimodal Reasoning via Functional Attention Control." arXiv preprint arXiv:2510.10285 (2025).

**Questions:**

The analysis appears to be based on single-token inference. Are the reported results averaged across all generated tokens for each answer, or do they correspond to specific tokens only?

---

> ### Author Response · Authors · 2025-11-20
> **Rebuttal Pt 1**
>
> **Dear Reviewer 3b67,**
>
> **We sincerely thank you for your careful review and thoughtful feedback! We have carefully revised the paper based on your suggestions. Please refer to the updated version, where all revisions and additions are highlighted in blue for your convenience.**
>
> Below we provide the point‑by‑point responses to all concerns (weaknesses and questions) raised in your review. **To facilitate your reading, we include TL;DR summaries ahead of detailed explanations whenever appropriate.**
>
> # Q1: Token used for getting VEA attention.
>
> > The analysis appears to be based on single-token inference. Are the reported results averaged across all generated tokens for each answer, or do they correspond to specific tokens only?
>
> ## Response
>
> **All reported results perform highlighting with first-token inference**, thus we do not need generate all tokens in the first pass. We detailed this in Appendix A.3 - VEA implementation details, and added this discussion in Section 3.2 - Attention Extraction to make it clearer.
>
> # W1: Modality shift from a complementry view of Figure 1
>
> > It remains unclear how a small increase in image RATP can be interpreted as a “modality shift,” especially when text attention is still orders of magnitude higher (e.g., 0.2 → 0.6 for image vs. ~20 for text). This weakens the claim that “vision plays a stronger role in later inference stages.”
>
> ## TL;DR
>
> **Thank you for the careful reading!**
>
> - We agree that “text attention is still orders of magnitude higher” is natural as the text tokens (in our case, the short question) are (i) far fewer (usually 5% or less) than image tokens, thus (ii) each token is carrying more dense information.
> - However, we note that the RAPT in Figure 1 is a **per-token** metric. **Our claim is more intuitive when considering total attention mass allocated to image versus text**: the total attention to image tokens increases from <1.0× of the text total in early layers to >2.5× in deeper layers. Which may more directly captures the modality shift we intend to highlight.
>
> **We are considering revise Fig. 1 to include this complementary analysis (or replace the current figure if preferred) to present the point more clearly, and we would also welcome your feedback on which presentation would be most helpful.**
>
> ## Additional Results
>
> The table below reports the ratio of total attention mass allocated to image tokens relative to text tokens across different layer ranges (e.g., 100% indicates equal total attention to image and text). All settings follow Figure 1. To highlight the overall trend, we group layers from shallow to deep into four intervals and report the mean within each group.
>
> | Layer Range | 0–8 | 9–16 | 17–24 | 25–32 |
> |-------------|-----|------|-------|-------|
> | Image Total Attention w.r.t. Text | 96.16% | 74.25% | 289.55% | 259.07% |
>
> Across these groups, the total attention to image tokens **increases from <1.0× of text in early layers to >2.5× in deeper layers**, showing a clear and consistent shift toward vision as inference progresses. This complementary view reinforces our modality shift claim and aligns with the trends discussed in Figure 1.

---

> ### Author Response · Authors · 2025-11-20
> **Rebuttal Pt 2**
>
> # W2: Discrepancies with existing findings
>
> > Prior studies (e.g., [1], [2]) have reported different attention trends, showing that image tokens already dominate attention in early layers. The authors should better situate their findings within this literature, clarifying methodological or model-related differences that explain these discrepancies.
>
> ## TL;DR
>
> Thank you for pointing us to these references. After examining both works carefully, we find that the discrepancies arise primarily from differences in emphasis and terminology rather than contradictory behavior. Our layer-wise trends align with existing findings once these distinctions are taken into account.
> **We have cited those suggested references and include relevant discussions in Appendix C.5.**
>
> ## Detailed Discussions
>
> - [1]'s main claim is that the overall **total attention** allocated to image tokens exceeds that of text tokens (Fig. 5 in [1]). **This matches our own findings and the complementary analysis we provided in our response to W1**: when averaged across layers, image tokens indeed receive more total attention than text tokens.
> - [2] states that “early layers show a high visual ratio with low concentration (broad scanning)”, **which aligns with our observation in Fig. 1 (very first few layers also have relatively high attention ratio) and Fig. 2 (those early layers have low concentration).** At the same time, it also emphasizes that “middle layers keep a high ratio while concentration rises (focusing and cross-modal alignment)”, **which also aligns with our findings as the “middle layers” in [2] correspond to what we refer to as “deeper layers” in our paper** (approximately the 50%–75% depth range), as shown in Fig. 1 of [2] and Fig. 3 of its cited source [3] published in CVPR 2025.
>
>
> > [1] Amara, Kenza, et al. "Why context matters in VQA and Reasoning: Semantic interventions for VLM input modalities." arXiv preprint arXiv:2410.01690 (2024).
> > [2] Lu, Haolang, et al. "Mitigating Hallucination in Multimodal Reasoning via Functional Attention Control." arXiv preprint arXiv:2510.10285 (2025).
> > [3] Bi, Jing, et al. "Unveiling visual perception in language models: An attention head analysis approach." Proceedings of the Computer Vision and Pattern Recognition Conference. 2025.
>
> # W3: Clarification on conclusion from Fig. 4
>
> > The interpretation of Fig. 4 could be refined: while the authors claim that evidence tokens receive higher attention even in incorrect answers, the plot also shows a drop in attention between correct and incorrect predictions, which complicates that conclusion.
> >
> ## Response
>
> Thank you for the careful reading and thoughtful comments. We would like to clarify that **our statement is only about whether the model, **when answering incorrectly**, still allocates *relatively higher* attention to evidence tokens.** This is what the dashed lines in Fig. 4 shows, and observing only the dashed lines is sufficient to support this point. Of course, we also agree that the model exhibits stronger grounding when it answers correctly, which is natural and intuitive, but this does not affect the logic of our conclusion regarding the incorrect cases. **We have revised the interpretation of Fig. 4 as you suggested to make this distinction clearer in the paper.**

---

> ### Author Response · Authors · 2025-11-20
> **Rebuttal Pt 3**
>
> # W4: Additional study on the reliability of deep-layer attention
>
> > The assumption that deep layers consistently capture “ground-truth” evidence regions is empirically plausible but not guaranteed. The paper would benefit from a clearer discussion on the reliability of these “evidence layers” across architectures and datasets.
>
> ## TL;DR
>
> Thank you for raising this important point. **Following you suggestion, we conduct additional quantitative and qualitative analyses to check (1) how often attention grounding is inaccurate, (2) how does VEA perform on such cases, and (3) what causes such grounding inaccuracies**. We confirmed that only a small fraction of samples (1.42%–7.34%) exhibit low grounding quality (AUROC <0.5), and for these cases, VEA still improves QA performance. This is because most low-AUROC cases stem from incomplete annotations rather than true grounding failures, and genuinely incorrect attention is rare and typically tied to vague or inconsistent dataset labels. We have incorporated these discussions into the paper at Appendix C.4 as suggested.
>
> ## Results and Discussions
>
> | Metric                             | TextVQA     | DocVQA      | SROIE       | InfographicsQA |
> |------------------------------------|-------------|-------------|-------------|----------------|
> | Low evidence AUROC (<0.5) Ratio    | 1.42%       | 3.11%       | 7.34%       | 5.20%          |
> | Base QA F1 score                   | 15.3        | 16.2        | 10.6        | 27.6           |
> | VEA QA F1 score                    | 21.8 (+6.5) | 26.4 (+10.2) | 18.9 (+8.3) | 35.3 (+7.7)   |
>
> - **Quantitative**: The Table above shows the ratio of low evidence AUROC (<0.5) samples in different dataset with Qwen-2.5VL-7B model. Across different datasets, only a small portion of samples (1.42% to 7.34%) exhibit low attention grounding AUROC. Importantly, for these samples, applying VEA highlighting still generally improves answer quality. The reasons are as follows.
> - **Qualitative**: We manually inspected samples with low evidence AUROC and found two main causes.
>     - (1) **Incomplete visual evidence annotation.** Many questions involve evidence that appears in multiple regions of the image, but only one region is labeled as evidence. The model attends to all relevant regions, which lowers AUROC despite the grounding being reasonable. In such cases, VEA still helps the model focus on the correct evidence.
>     - (2) **When the model fails to perceive the evidence entirely and instead attends to broad irrelevant areas, the failures often arise from vague or imperfect question or answer annotations.** e.g., the question ask for an aircraft identification number while the image contains only the aircraft model. In such cases, the model has no accessible evidence to ground upon, and VEA, which is designed to help the model focus on perceived evidence, cannot compensate for missing or incorrect data annotations.

---

> ### Author Response · Authors · 2025-11-20
> **Rebuttal Pt 4**
>
> # W5: On the purpose and robustness of evidence-layer selection
>
> > It is also unclear whether the identification of such evidence layers must be repeated per model, and how sensitive the steering mechanism is to inaccuracies in this identification.
>
> ## TL;DR
>
> * *Purpose of profiling*:
>   **Model specific evidence layer profiling helps further improve the robustness of highlighting, since different models show slightly different layer positions where evidence grounding is strongest.** This can be seen in Fig. 4, and we provide additional discussion and visualizations in Appendix C.3 (Fig. 10). Profiling ensures that VEA steers the model at layers that naturally exhibit stronger visual evidence awareness.
> * *Sensitivity to inaccurate layer selection*:
>   **VEA can still get competitive performance with simple static layer choice.** As shown in Table 2, using the later 50% of layers without profiling still yields competitive evidence identification accuracy that is close to other baselines such as AGLA. Profiling, however, can further improve accuracy, as reflected in Tables 2 and 3. **We further test the QA performance of multiple static (thus inaccurate) layer choices to confirm the robustness of VEA to inaccurate layer identification, as detailed below.**
>
> **We have include relevant results and analysis in Appendix C.6.**
>
> ## Detailed Results and Discussions
>
> > Whether the identification of such evidence layers must be repeated per model?
>
> | Evidence AUROC | llava 7B | llava 13B | qwen 7B | qwen 32B | gemma 4B | gemma 27B |
> |-|-|-|-|-|-|-|
> | L_{0\%-100\%}     | 75.9  | 76.3  | 68.5  | 57.0  | 59.5  | 61.8  |
> | L_{0\%-50\%}      | 68.2  | 73.1  | 59.4  | 51.3  | 56.5  | 55.4  |
> | L_{50\%-100\%}    | 78.0  | 76.9  | 79.5  | 67.6  | 65.9  | 68.1  |
> | L_{50\%-75\%}     | ***81.2*** | ***79.1*** | ***80.3*** | 65.9  | ***71.7*** | ***70.0*** |
> | L_{75\%-100\%}    | 74.8  | 74.7  | 78.7  | ***70.4*** | 60.8  | 66.3  |
> | VEA Profiling       | **83.6** | **84.4** | **85.2** | **79.1** | **80.0** | **81.2** |
>
> The table above reports the mean evidence attribution AUROC on four datasets for different static layer choices. In addition to Table 2, we include two more ranges L_{50\%-75\%} and L_{75\%-100\%}. We observe that the static choice of L_{50\%-75\%} still achieves strong attribution quality (75.9–90.8 AUROC across six models). Profiling each model, however, consistently yields more robust results (88–92.2 AUROC).
>
> Profiling’s advantage is especially clear for the Gemma family which exhibit a periodic pattern that “good attribution layers” appear intermittently rather than in a continuous block (Appendix C.3, Fig. 9). Static continuous ranges fail to capture this behavior, highlighting the benefit of per model profiling.
>
> > How sensitive the steering mechanism is to inaccuracies in this identification?
>
> | QA F1 Score        | llava 7B | llava 13B | qwen 7B | qwen 32B | gemma 4B | gemma 27B |
> |-|-|-|-|-|-|-|
> | Base               | 27.8  | 66.8  | 79.5  | 72.1  | 58.2  | 57.6  |
> | L_{0\%-100\%}    | 58.2  | 72.8  | 86.6  | 78.2  | 64.5  | 61.2  |
> | L_{0\%-50\%}     | 38.9  | 71.6  | 80.1  | 71.8  | 59.9  | 56.4  |
> | L_{50\%-100\%}   | 62.2  | 73.1  | ***87.6*** | 79.8  | 65.4  | 62.0  |
> | L_{50\%-75\%}    | ***66.7*** | ***74.1*** | 87.5 | 78.8  | ***66.9*** | ***62.1*** |
> | L_{75\%-100\%}   | 56.5  | 71.9  | 87.1 | ***80.6*** | 63.6  | ***62.1*** |
> | VEA Profiling      | **69.4** | **76.3** | **88.6** | **81.4** | **69.6** | **63.2** |
>
> We further evaluate QA performance on TextVQA using multiple static (and thus intentionally inaccurate) layer choices. VEA with static middle or deep layers (later 50%) still delivers strong gains across model families, **showing that VEA is robust to suboptimal layer identification, but progfiling still offers additional improvements.**
>
> We also observe that early layers generally lack visual grounding behavior. Consequently, using L_{0\%-50\%} yields mixed results and sometimes degrades performance, especially for Qwen and Gemma models. This further supports our conclusion: visual grounding is not uniformly present across layers, and selecting middle or deep layers, or using model specific profiling, is important for accurate evidence highlighting.
>
> # Happy to have further discussion!
>
> **Thank you again for your thoughtful review. With the valuable feedback from you and the other reviewers, we’ve put significant effort into generating new results and expanding our analysis accordingly. We hope our responses have addressed your concerns, and we would be happy to discuss further if you have any additional questions.**

---

> ### Author Response · Authors · 2025-11-27
> **Looking forward to your reply : )**
>
> Dear Reviewer 3b67,
>
> **Thank you again for taking the time to review our work! We appreciate your detailed feedback and the concerns you raised : )**
>
> As we are now midway through the discussion phase, we wanted to gently check in and see whether our earlier responses and the updated manuscript have addressed your points. If any part would benefit from further clarification, we would be glad to elaborate.
>
> We also understand that this requires additional time, but if you have a moment, we would be grateful if you could take another look at the revised version and the six pages of new experiments and analysis. For your convenience, we made the following revisions based on your comments:
> - Q1: We have revised Section 3.2 to clarify the settings better.
> - W1: We provide complementary analysis in the rebuttal, and would appreciate your feedback on the best way to present it in Figure 1.
> - W2: We have cited those suggested references and include relevant discussions in Appendix C.5. In short, we find that the discrepancies arise primarily from differences in emphasis and terminology rather than contradictory behavior.
> - W3: We have revised the interpretation of Fig. 4 as you suggested.
> - W4: Discussions are in Appendix C.4 - When attention is not reliable.
> - W5: Please see relevant results and analysis in Appendix C.6 - Evaluation on the effect of evidence-layer selection.
>
> **Your feedback has meaningfully improved the quality of our paper, and we are sincerely grateful for your time, effort, and any possible reconsideration. Looking forward to your reply!**
>
> Best regards,
> Authors

---

### Official Review · Reviewer_8FW4 · 2025-10-24

**Soundness:** 3
**Presentation:** 3
**Contribution:** 3
**Rating:** 6
**Confidence:** 3

**Summary:**

The paper proposes a phenomenon called “seeing but not believing” by studying the internal layers of vision-language models (VLMs). Through analyses of layer-wise attention dynamics and layer-wise profiling, the authors introduce a visual evidence augmentation approach that enhances regions likely to contain stronger visual evidence. The proposed method is training-free and provides both improved interpretability and better performance for VLMs.

**Strengths:**

1. The paper is well written and easy to follow. The phenomenon of “seeing but not believing” is intriguing.

2. The experiments and ablation studies are comprehensive.

3. The proposed VEA approach provides both interpretability and performance improvement.

**Weaknesses:**

1. Some experimental details are missing. For example, in Figures 2 and 3, it is unclear which models were used for attention map visualization.

2. Including more visual examples could strengthen and better support the overall narrative.

**Questions:**

Did the authors observe the attention sink phenomenon [1,2] during their experiments? If so, how did they handle these situations?

Reference:
[1] See What You Are Told: Visual Attention Sink in Large Multimodal Models, ICLR 25
[2] Vision Transformers Need Registers, ICLR 24

---

> ### Author Response · Authors · 2025-11-20
> **Rebuttal**
>
> **Dear Reviewer 8FW4,**
>
> **We sincerely thank you for your valuable feedback and for recognizing the strengths of our work! We have carefully revised the paper based on your suggestions. Please refer to the updated version, where all revisions and additions are highlighted in blue for your convenience.**
>
> Below we provide the point‑by‑point responses to all concerns (weaknesses and questions) raised in your review. **To facilitate your reading, we include TL;DR summaries ahead of detailed explanations whenever appropriate.**
>
> # Q1: On the attention sink phenomenon.
>
> > Did the authors observe the attention sink phenomenon [1,2] during their experiments? If so, how did they handle these situations?
>
> ## TL;DR
>
> **Yes, we did observe the attention sink phenomenon. The attention mask denoising step in our method is specifically designed to address this issue.** We have revised the paper to improve clarity by adding related discussion to Figure 2 (visualization of image attention across different layers) and citing suggested references in Section 3.2.
>
> ## Response
>
> Thank you for pointing this out. We indeed observed the vision/attention sink phenomenon during our experiments. As discussed in Section 3.2 on attention mask denoising, Figures 2 and 3 show that raw attention masks often contain scattered high-valued patches unrelated to true evidence. However, this does not affect the overall grounding behavior. We also find that, unlike genuine evidence regions that form spatially coherent clusters tied to the query, sink-driven patches appear as isolated artifacts. To suppress these sink tokens, we introduce a neighborhood-based filtering step to remove spurious activations and stabilize the visual attention maps (please refer to Section 3.2 for more details).
>
> # W1: Setup details of Figure 2 and 3
>
> > Some experimental details are missing. For example, in Figures 2 and 3, it is unclear which models were used for attention map visualization.
>
> ## Response
>
> **Thank you for the careful reading. Figures 2 and 3 follow the same setup as Figure 1 and use the LLaVA-1.5-7B model for visualization. We have revised the figure captions to clarify these experimental details.**
>
> # W2: More visual examples
>
> > Including more visual examples could strengthen and better support the overall narrative.
>
> ## Response
>
> **Thank you for the feedback. We have added additional qualitative examples in the Figure 8.** To further enrich scenario diversity, we included extra natural examples from the MMStar dataset in addition to the four main datasets used in the paper. Please refer to Appendix C.7 and Figure 8 (on page 22) in the updated version for more details.
>
>
> # Happy to have further discussion!
>
> **Thank you again for your thoughtful review. With the valuable feedback from you and the other reviewers, we’ve put significant effort into generating new results and expanding our analysis accordingly. We hope our responses have addressed your concerns, and we would be happy to discuss further if you have any additional questions.**

---

> > ### Comment · Reviewer_8FW4 · 2025-11-27
> >
> > thanks authors for the clarification, I will maintain my original rating.

---

> > > ### Author Response · Authors · 2025-11-27
> > >
> > > Dear Reviewer 8FW4,
> > >
> > > Thank you for the response! Please don't hesitate to let us know if you have any further questions, we would be glad to answer them.
> > >
> > > Thanks,
> > > Authors

---

### Official Review · Reviewer_LvsH · 2025-11-01

**Soundness:** 3
**Presentation:** 3
**Contribution:** 2
**Rating:** 6
**Confidence:** 3

**Summary:**

This work investigates the phenomenon of “seeing and not believing”, where vision-language models (VLMs) attend to the correct visual regions but still produce incorrect answers. To address this issue, the authors propose an inference-time method that encourages VLMs to better leverage visual information by overlaying attention-derived masks on the input image. Experiments across four VQA benchmarks and four VLM families demonstrate the effectiveness of the proposed approach.

**Strengths:**

1. The authors conduct a thorough investigation of how different layers in VLMs process inputs and distribute attention, providing valuable insights for the community.
2. The proposed solution is simple yet effective, showing consistent  improvements across eight different VLMs.

**Weaknesses:**

1. It is important to evaluate cases where the model does not attend to the correct regions.
How often does the model still answer correctly in such cases?
Does performance degrade when highlighting regions based on incorrect attention?
2. The motivation is somewhat similar to [1], which also identifies this phenomenon and proposes attention-based approaches.
This overlap reduces the novelty of the contribution, though the improvements of the method are still appreciated.


[1] "Unveiling the Ignorance of MLLMs: Seeing Clearly, Answering Incorrectly", Liu et al.

**Questions:**

1. In cases where the model attends to incorrect regions but still answers correctly, what happens when the attention-derived mask is applied?
Does it lead to performance degradation in such examples?
2. I would be interested to see qualitative examples that explore attention behavior in more general, real-world scenes, beyond the text-centric scenarios that are (to my understanding) the main focus of the examined datasets.

---

> ### Author Response · Authors · 2025-11-20
> **Rebuttal**
>
> **Dear Reviewer LvsH,**
>
> **We sincerely thank you for your valuable feedback and for recognizing the strengths of our work! We have carefully revised the paper based on your suggestions and incorporated additional complementary experiments and analyses. Please refer to the updated version, where all revisions and additions are highlighted in blue for your convenience.**
>
> Below we provide the point‑by‑point responses to all concerns (weaknesses and questions) raised in your review. **To facilitate your reading, we include TL;DR summaries ahead of detailed explanations whenever appropriate.**
>
>
> # Q1 & W1: When the attention is not reliable.
>
> > It is important to evaluate cases where the model does not attend to the correct regions. How often does the model still answer correctly in such cases? Does performance degrade when highlighting regions based on incorrect attention?
>
> ## TL;DR
>
> Thank you for raising this important point. **Following you suggestion, we conduct additional quantitative and qualitative analyses to check (1) how often attention grounding is inaccurate, (2) how does VEA perform on such cases, and (3) what causes such grounding inaccuracies**. We confirmed that only a small fraction of samples (1.42%–7.34%) exhibit low grounding quality (AUROC <0.5), and for these cases, VEA still improves QA performance. This is because most low-AUROC cases stem from incomplete annotations rather than true grounding failures, and genuinely incorrect attention is rare and typically tied to vague or inconsistent dataset labels. We have incorporated these discussions into the paper at Appendix C.4 as suggested.
>
>
> ## Results and Discussions
>
> | Metric  | TextVQA     | DocVQA      | SROIE       | InfographicsQA |
> |---|----|--|----|-|
> | Low evidence AUROC (<0.5) Ratio    | 1.42%       | 3.11%       | 7.34%       | 5.20%          |
> | Base QA F1 score | 15.3        | 16.2        | 10.6        | 27.6           |
> | VEA QA F1 score | 21.8 (+6.5) | 26.4 (+10.2) | 18.9 (+8.3) | 35.3 (+7.7)   |
>
> - **Quantitative**: The Table above shows the ratio of low evidence AUROC (<0.5) samples in different dataset with Qwen-2.5VL-7B model. Across different datasets, only a small portion of samples (1.42% to 7.34%) exhibit low attention grounding AUROC. Importantly, for these samples, applying VEA highlighting still generally improves answer quality. The reasons are as follows.
> - **Qualitative**: We manually inspected samples with low evidence AUROC and found two main causes.
>     - (1) **Incomplete visual evidence annotation.** Many questions involve evidence that appears in multiple regions of the image, but only one region is labeled as evidence. The model attends to all relevant regions, which lowers AUROC despite the grounding being reasonable. In such cases, VEA still helps the model focus on the correct evidence.
>     - (2) **The model fails to perceive the evidence entirely and instead attends to broad irrelevant areas. These failures often arise from vague or imperfect question or answer annotations.** e.g., the question ask for an aircraft identification number while the image contains only the aircraft model. In such cases, the model has no accessible evidence to ground upon, and VEA, which is designed to help the model focus on perceived evidence, cannot compensate for missing or incorrect data annotations.
>
>
> # Q2: Qualitive examples in natural scenes.
>
> > I would be interested to see qualitative examples that explore attention behavior in more general, real-world scenes, beyond the text-centric scenarios that are (to my understanding) the main focus of the examined datasets.
>
> ## Response
>
> Thank you for the feedback. We have added additional qualitative examples in the appendix. Following your suggestion, beyond the four original datasets, we included non-text-centric scenarios from the MMStar dataset to increase the diversity of examples. **Please refer to Appendix C.7 and Figure 8 (on page 22) in the updated version of the paper.**
>
> # W2: Relationship to [1]
>
> > The motivation is somewhat similar to [1], which also identifies this phenomenon and proposes attention-based approaches. This overlap reduces the novelty of the contribution, though the improvements of the method are still appreciated.
>
> ## Response
>
> Thank you for your careful reading. As described in paragraphs 2–3 of the introduction, our work is inspired by [1], which also serves as one of our main baselines. **However, while they focuses on a benchmark-oriented investigation, our paper takes a different perspective by conducting a layer-wise attention analysis to better understand the internal mechanisms behind this phenomenon.** Through this deeper analysis, we uncover several novel findings about attention dynamics. The proposed method is a natural extension of these insights, incorporating additional denoising and smoothing mechanisms, which lead to improved performance over [1].

---

> ### Author Response · Authors · 2025-11-20
> **Happy to have further discussion!**
>
> # Happy to have further discussion!
>
> **Thank you again for your thoughtful review. With the valuable feedback from you and the other reviewers, we’ve put significant effort into generating new results and expanding our analysis accordingly. We hope our responses have addressed your concerns, and we would be happy to discuss further if you have any additional questions.**

---

> ### Author Response · Authors · 2025-11-27
> **Looking forward to your reply : )**
>
> Dear Reviewer LvsH,
>
> **Thank you again for your thoughtful and supportive review. We truly appreciate your positive assessment of our work.** As we are now midway through the discussion phase, we wanted to gently check in and see whether our earlier clarifications and the updated manuscript have fully addressed your comments. If anything could benefit from further explanation, we would be happy to elaborate.
>
> We also understand that this takes additional time, but if you have a moment, we would be grateful if you could take a brief look at the revised version and the six pages of new experiments and analysis, just to ensure everything aligns with your expectations. For your convinience, we made following revisions based on your feedback:
> - Q1 & W1: Appendix C.4 - When attention is not reliable
> - Q2: Appendix C.7 - More visual examples, and Figure 8
>
> **Your feedback has helped us strengthen the paper, and we sincerely appreciate the time and care you have invested.**
>
> Best regards,
> Authors

---

### Official Review · Reviewer_RdjA · 2025-11-01

**Soundness:** 4
**Presentation:** 3
**Contribution:** 3
**Rating:** 6
**Confidence:** 4

**Summary:**

This paper focuses on the "seeing but not believing" phenomenon in Vision-Language Models (VLMs): VLMs often perceive correct visual evidence in Visual Question Answering (VQA) but fail to leverage it for accurate answers. Through layer-wise attention analysis, it reveals three key findings: shallow VLMs layers focus on text, deeper layers sparsely attend to localized evidence regions, and deeper layers still lock onto evidence even when outputs are wrong. To solve this, the paper proposes VEA (Visual Evidence Augmentation), a training-free inference-time method. It first identifies "visual-grounding layers". During inference, VEA extracts attention from these layers, applies denoising and Gaussian smoothing to create an evidence mask, and fuses the mask with the original image to highlight evidence (weakening non-evidence regions).

**Strengths:**

1. The paper presents an insightful observation on VLMs’ behaviors toward images. Its visualizations and analyses further reveal how text and image interactions are modeled across different layers, showing that the encoding of semantic features first emerges in deeper layers. The inconsistency between attention maps (i.e., "seeing but not believing") highlights limitations of current VLM architectures, which is valuable for guiding future research.

2. To address this issue, the authors propose a simple yet effective algorithm. The design is straightforward and practical: it can be applied to various VLMs with zero training cost and demonstrates effectiveness across multiple benchmarks.

**Weaknesses:**

1. The method improves VLM performance by overlaying a salient mask on the input image, but it does not "fix the VLM’s attention behavior" (as the behavior of the VLM or attention is not changed). Additionally, the design of the algorithm will introduce extra cost, and also raises the convern about multi-turn/multi-image scenarios.

2. The proposed algorithm augments the brightness of different regions of the image. However, this augmentation will change the original image, causing information loss and changes (e.g., this will influence questions about brightness or color). For questions that rely on global context or beyond retrieval, the proposed mask method might cause troubles.

3. Compared with visual reasoning methods that interactively retrieve key parts of the image to gather information, the proposed approach is plug-and-play, however, it also introduce limitations. This point could be further discussed in the paper.

**Questions:**

1. The paper proposes to improve the network attention by casting masks on the input image. Why not apply this method to intermediate features or attention?
2. How does this method work on multiturn conversations, multi-image QA or video?3.
3. How does this method influence the benchmarks that rely on global context or multiple elements, such as benchmarks for general knowledge understanding or math (such as MMMU, MMStar, AI2D, MathVista)?
4. As the author suggested, the answer from a single inference might not be accurate. The same applies to the attention map — can this method apply to the same question and image multiple times in a cascade way?

---

> ### Author Response · Authors · 2025-11-20
> **Rebuttal (Pt 1)**
>
> **Dear Reviewer RdjA,**
>
> **We sincerely thank you for your valuable feedback and for recognizing the strengths of our work! We have carefully revised the paper based on your suggestions and incorporated additional complementary experiments and analyses. Please refer to the updated version, where all revisions and additions are highlighted in blue for your convenience.**
>
> Below we provide the point‑by‑point responses to all concerns (weaknesses and questions) raised in your review. **To facilitate your reading, we include TL;DR summaries ahead of detailed explanations whenever appropriate.**
>
> # Q1 & W1: Why not model-level intervention?
>
> > The paper proposes to improve the network attention by casting masks on the input image. Why not apply this method to intermediate features or attention?
>
> ## TL;DR
>
> Thank you for the thoughtful suggestion. **We actually explored this direction in the early stage of the project, but found that intervening in the model’s internal reasoning process can make the output unstable.** Small interventions had negligible effects, while stronger interventions often caused severe output degradation. **After weighing the trade-offs, we chose to enhance the input image instead, which proved to be more stable, general, and capable of achieving similar effects with a much simpler mechanism.**
>
> ## Response
>
> In the early stage of this project, inspired by works such as \[1,2\], we initially attempted to enhance visual evidence utilization by directly intervening in the model’s inference process. Specifically, by amplifying the attention weights of selected layers/heads (using the profiling strategy in Section 3.1 for selection) toward visual evidence tokens at certain intermediate layers.
>
> However, we observed that such interventions affected the reasoning process in highly unpredictable ways, often leading to generation collapse, and the stronger the intervention, the more frequent the collapse. **We tested on the LLaVA and Qwen model families, and found that modifying intermediate attention can cause LLaVA to repeatedly output sequences like “!!!!!” after a few tokens, while Qwen models tended to insert special tokens such as “\naddCriterion” and terminate generation abruptly (this is also reported by others [link](https://github.com/QwenLM/Qwen3-VL/issues/759)), or even mix Chinese text into English responses.** Reducing the intervention strength made the model output stable again, but also with no observable help in evidence utilization. **Moreover, even under the same intervention strength, some samples remained unaffected while others faces generation failures.**
>
> While it might be possible to design a complex adaptive intervention mechanism to mitigate these issues, we decided to follow the principle of avoiding unnecessary complexity. We therefore adopted an input-level enhancement approach, which improves evidence utilization while remaining simple, stable, and model-agnostic.

---

> ### Author Response · Authors · 2025-11-20
> **Rebuttal (Pt 2)**
>
> # Q2 & W1: Multi-turn or multi-image setting.
>
> > How does this method work on multiturn conversations, multi-image QA or video?
>
> ## TL;DR
>
> **Following your suggestion, we extended our experiments to multi-turn conversations and multi-image QA tasks. The results show that VEA consistently improves visual evidence utilization across different models in both scenarios.** As for video-based tasks, we did not include them in this work since the underlying architectures and processing pipelines differ significantly from the scope of our current study. We have added relevant results and analysis in Appendix C.1 - Additional Results and Analysis.
>
> ## Detailed Results and Analysis
>
> **Multi-turn QA:**
> For the multi-turn QA setting, we evaluated VEA on the Visual Dialog (VisDial) dataset \[1\] to test whether it can consistently enhance model performance across rounds of image-grounded dialogue. Because the VisDial answers are free-form natural language, we use Token F1 and Exact Match (EM) as metrics. For each dialogue turn, we include the entire dialog history in the text context. VEA is applied based on the attention from the first decoding step of the current question.
>
> | VisDial  | LLaVA-1.6-7B | Qwen2.5VL-7B | Gemma3-4B | InternVL3.5-8B |
> |:----------|:--------------|:--------------|:------------|:----------------|
> | Base F1  | 28.5 | 27.5 | 25.0 | 33.2 |
> | VEA F1   | 44.9 (+16.4) | 47.8 (+20.3) | 41.2 (+16.2) | 48.6 (+15.4) |
> | Base EM  | 38.4 | 37.6 | 34.0 | 38.5 |
> | VEA EM   | 40.9 (+2.5) | 40.5 (+2.9) | 37.5 (+3.5) | 41.4 (+2.9) |
>
> **Analysis:**
> VEA consistently improves both F1 and EM across models, with larger gains on F1. This is because VisDial answers are free-form phrases rather than fixed entities directly retrievable from images. Hence, semantically plausible answers may not achieve EM matches but still show improvements in token-level overlap. For example, if the ground truth answer is *“they are skiing”*, the base model might answer *“can’t tell”*, while VEA outputs *“skiing”*. Both fail EM, but the latter achieves higher Token F1, which we find to be a more informative measure in this setting.
>
>
> **Multi-image QA:**
> We further evaluated VEA on the BLINK dataset \[2\], which focuses on multi-image reasoning beyond textual descriptions. Among its subtasks, we selected the Semantic Correspondence task, which requires the model to reason over multiple detailed visual cues across images. Since the task is multiple-choice, we use Accuracy to measure performance.
>
> | BLINK   | LLaVA-1.6-7B | Qwen2.5VL-7B | Gemma3-4B | InternVL3.5-8B |
> |:----------|:--------------|:--------------|:------------|:----------------|
> | Base ACC | 34.4 | 61.4 | 42.2 | 66.4 |
> | VEA ACC  | 45.2 (+10.8) | 68.9 (+7.5) | 50.9 (+8.7) | 71.3 (+4.9) |
>
> **Analysis:**
> VEA also provides consistent improvements on the multi-image QA task. Later model like InternVL3.5 already demonstrates strong multi-image reasoning ability, but VEA can further enhances its performance. This confirms that emphasizing visual evidence via VEA benefits both conversational and multi-image reasoning scenarios.
>
> > [1] Das, Abhishek, et al. "Visual dialog." Proceedings of the IEEE conference on computer vision and pattern recognition. 2017.
> > [2] Fu, Xingyu, et al. "Blink: Multimodal large language models can see but not perceive." European Conference on Computer Vision. Cham: Springer Nature Switzerland, 2024.

---

> ### Author Response · Authors · 2025-11-20
> **Rebuttal (Pt 3)**
>
> # Q3 & W2: Will VEA hinder global context understanding?
>
> > How does this method influence the benchmarks that rely on global context or multiple elements, such as benchmarks for general knowledge understanding or math (such as MMMU, MMStar, AI2D, MathVista)?
>
> ## TL;DR
>
> - **The datasets we used also rely on global context:** For example, InfographicsQA requires counting, sorting, and connecting multiple elements across the image to obtain the correct answer.
> - **When VEA may affect global context understanding:** As discussed in RQ4 (Fig. 7), excessively strong intervention strength (e.g., $\alpha=1$) can indeed mask too much background, hindering global visual context understanding. However, within a wide and reasonable range ($\alpha \in [0.25, 0.75]$), VEA consistently outperforms the baseline, indicating that emphasizing evidence regions does not compromise global context comprehension.
> - **Additional results on new datasets:** Following your suggestion, we evaluated VEA on MMStar and AI2D, where MMStar also includes samples from MMMU and MathVista. Results show that VEA consistently improves model performance on these reasoning-centric benchmarks. Inspired by your suggestion, we further tested a variant VEA\* that retains both the original and highlighted images, which further enhances global context understanding. These results and discussions are also included in Appendix C.2.
>
> ## Additional Results and Analysis
>
> Following your suggestion, we conducted experiments on MMStar and AI2D. These datasets emphasize reasoning and domain knowledge rather than visual perception alone. They typically involve large and clear visual elements, meaning visual evidence utilization is not the primary challenge.
>
> To better preserve global context, we also tested a variant VEA\* that feeds both the original and highlighted images to the model. Since both datasets are multiple-choice tasks, we report accuracy as the evaluation metric.
>
> | AI2D   | LLaVA-1.6-7B | Qwen2.5VL-7B | Gemma3-4B | InternVL3.5-8B |
> |:---------|:-------------|:-------------|:-----------|:----------------|
> | Base ACC | 66.4 | 78.9 | 56.2 | 77.2 |
> | VEA ACC  | 69.8 (+3.4) | 81.1 (+2.2) | 61.5 (+5.3) | 81.8 (+4.6) |
> | VEA* ACC | 70.2 (+3.8) | 82.3 (+3.4) | 63.3 (+7.1) | 83.7 (+6.5) |
>
> | MMStar   | LLaVA-1.6-7B | Qwen2.5VL-7B | Gemma3-4B | InternVL3.5-8B |
> |:-----------|:-------------|:-------------|:-----------|:----------------|
> | Base ACC | 36.7 | 48.4 | 32.8 | 53.7 |
> | VEA ACC  | 40.5 (+3.8) | 52.8 (+4.4) | 37.5 (+4.7) | 57.6 (+3.9) |
> | VEA* ACC | 40.2 (+3.5) | 53.3 (+4.9) | 39.1 (+6.3) | 58.2 (+4.5) |
>
> **Analysis:**
> - VEA consistently improves performance on both benchmarks, though the magnitude of improvement is smaller compared with datasets that rely more heavily on fine-grained visual evidence. This aligns with the fact that these reasoning tasks emphasize logical inference rather than evidence localization.
> - The VEA\* variant (retaining both the original and highlighted images) further enhances accuracy by helping the model maintain access to the global context. However, this approach is more effective for models with strong multi-image reasoning capability (e.g., Qwen2.5-VL, InternVL). For models not trained on multi-image inputs (e.g., LLaVA-1.6-7B), VEA\* yields limited additional benefit due to weaker multi-image reasoning ability.

---

> ### Author Response · Authors · 2025-11-20
> **Rebuttal (Pt 4)**
>
> # Q4: VEA with multi-round self-validation.
>
> > As the author suggested, the answer from a single inference might not be accurate. The same applies to the attention map — can this method apply to the same question and image multiple times in a cascade way?
>
> ## TL;DR
>
> This is an excellent question. **Inspired by your suggestion, we explored whether introducing a cascaded self-validation mechanism could further improve VEA.** Specifically, we designed a multi-round self-validation pipeline, where after generating an initial answer, the model repeatedly validates and re-applies VEA to obtain updated highlighted images. **However, the results show that this strategy brings only marginal gains: models tend to agree with their previous answers, indicating that current VLMs still have limited self-reflection ability.** We have added relevant results and analysis in Appendix C.3.
>
> ## Additional Results and Discussion
>
> **Setup:** We implemented a variant called VEA-cascade. After generating the initial VEA response, we performed multiple rounds of self-validation. In each round, the model receives the previously highlighted image and question, and is asked to assess whether its last answer was correct. If incorrect, the model produces a new answer and re-applies VEA based on the updated attention map. If correct, the model outputs “[I confirm this answer is correct]”, and we take the previous answer as final. We set the maximum number of validation rounds to 3 and evaluated the approach on Qwen2.5VL-7B. The results are shown below.
>
> | Method | TextVQA | DocVQA | SROIE | InfographicsQA |
> |:--------|:-----------|:-----------|:-----------|:----------------|
> | VEA EM | 90.3 | 76.2 | 82.5 | 64.4 |
> | VEA-cascade EM | 90.3 (+0.0) | 76.6 (+0.4) | 82.7 (+0.2) | 64.4 (+0.5) |
> | Avg. #Rounds | 1.05 | 1.09 | 1.10 | 1.07 |
>
> **Analysis:**
> - The multi-round self-validation mechanism does not yield significant improvement. We also observe that the average number of reflection rounds remains close to 1, meaning that the VLM almost always confirms its previous answer. The number of reflection rounds is slightly lower on simpler datasets such as TextVQA but overall differences remain minor.
> - This suggests that non-reasoning-oriented VLMs possess limited self-reflection capabilities, and more sophisticated mechanisms are needed to effectively prompt genuine self-correction.
> - Inspired by recent studies on VLM reflection [1,2], we believe that although this simple cascaded approach shows limited benefit, leveraging VEA’s attention insights to enable dynamic reflection during long reasoning processes, possibly with real-time attention tracking and image editing tools, could be a promising direction for future work. We also discuss this idea in our response to W3.
>
> > [1] Wei, Yana, et al. "Perception in reflection." arXiv preprint arXiv:2504.07165 (2025).
> > [2] Jiang, Chaoya, et al. "VLM-R $^ 3$: Region Recognition, Reasoning, and Refinement for Enhanced Multimodal Chain-of-Thought." arXiv preprint arXiv:2505.16192 (2025).
>
> # W3: About limitations of VEA.
>
> > Compared with visual reasoning methods that interactively retrieve key parts of the image to gather information, the proposed approach is plug-and-play, however, it also introduce limitations. This point could be further discussed in the paper.
>
> ## Response
>
> Thank you for your suggestion. We have discussed the limitations and potential solutions of VEA in Appendix B.2. We agree that extending VEA toward a more interactive paradigm (where the model dynamically retrieves key image regions based on its current reasoning step and state) would be an exciting direction. This does not necessarily require complex reasoning-based or reinforcement learning–based fine-tuning with high-quality data; the model’s internal attention signals could potentially offer/supplement this capability. For example, during reflection, VEA with various image editing (e.g., cropping/denoising/contrasting) could be used to help the model dynamically focus only on the critical regions of the original image, thereby mitigating the “ineffective reflection” phenomenon reported in prior literature [1]. Please also kindly refer to Appendix B.3 for more relevant discussions.
>
> > [1] Zhang, Wenqi, et al. "Self-Contrast: Better Reflection Through Inconsistent Solving Perspectives." Proceedings of the 62nd Annual Meeting of the Association for Computational Linguistics (Volume 1: Long Papers). 2024.
>
> # Happy to have further discussion!
>
> **Thank you again for your thoughtful review. With the valuable feedback from you and the other reviewers, we’ve put significant effort into generating new results and expanding our analysis accordingly. We hope our responses have addressed your concerns, and we would be happy to discuss further if you have any additional questions.**

---

> > ### Comment · Reviewer_RdjA · 2025-11-26
> > **replay to authors**
> >
> > Thanks for your detailed response. I have read it carefully, along with the other reviews, and most of my concerns have been resolved. As I'm mostly positive about this paper.

---

> > > ### Author Response · Authors · 2025-11-26
> > > **Thank you for the response!**
> > >
> > > Dear Reviewer RdjA,
> > >
> > > Thank you for your response! We are glad to hear that our rebuttal helped address your concerns : )
> > >
> > > **We understand that this will take extra of your valuable time, but may we kindly ask if you would be willing to reconsider your evaluation in light of the revised version and the ~6 pages of new experiments/analysis?**
> > >
> > > **Your feedback has greatly contributed to further improving our paper, and we are sincerely grateful for your time, effort, and possible reconsideration in this review. Thank you again!**
> > >
> > > With best regards,
> > > Authors

---

### Public Comment · ~Jiarui_Zhang2 · 2025-11-12
**A highly similar work**

Dear authors,

I would like to draw your attention to a highly similar and related work that was not cited in the current paper: “MLLMs Know Where to Look: Training-Free Perception of Small Visual Details with Multimodal LLMs,” published at ICLR 2025.

In particular, the analysis presented in RQ3 of your paper is nearly identical to Section 4 of our work, covering both the qualitative and quantitative analyses, which represent key motivations and findings in both studies. Moreover, the visual intervention approach shown in Figure 5 is also similar to our Vicrop method: both leverage the MLLM’s internal attention from the initial inference step to guide visual information enhancement, after which the refined visual input is reintroduced into the model, while we recognize the differences in implementation details, such as the highlighting strategies (cropping vs. alpha blending) and attention extraction techniques (relative vs. neighborhood filtering).

Given these conceptual and methodological similarities, I would appreciate it if you could acknowledge and discuss our work in the relevant sections.

Thank you

---

> ### Author Response · Authors · 2025-11-20
> **Thanks but they are NOT "highly similar"**
>
> Hi Jiarui,
>
> Thank you for your interest and bringing your work to our attention! **However, after careful reading and to our best understanding, we feel *it is not accurate to describe the two papers as “highly similar”* (which usually refers to significant overlap in motivation, analysis, and methodology) for the following reasons, but we welcome any further questions or thoughts from your side:**
>
> ## 1. Motivation
>
> As discussed in paragraphs 2–4 of ViCrop’s introduction, your main motivation is that **the size of objects in the image affects VLM perception**, and you further analyze how different cropping strategies influence perception in Fig. 1 and Section 3. Our work, on the other hand, starts from a layer-wise attention analysis inside the model, exploring **why the model fails to leverage the visual evidence it has already perceived**. As we discussed in paragraph 3 of our introduction, our motivation is inspired by existing literature \[1,2,3\].**
>
> ## 2. Analysis
>
> **ViCrop's analysis (before Section 4) focus on how cropping affects model perception.** Fig. 1 examines the influence of cropping on output probabilities. Section 3 (Table 1) then provides further quantitative verification that appropriate cropping improves model performance. **VEA's analysis are all concentrate on layer-wise attention analysis**: Section 2.1 investigates modality attention mass transition across layers, Section 2.2 examines the transition of image-attention patterns from shallow to deep layers, and Section 2.4 discusses why a model might perceive but ignore visual evidence.
>
> Your analysis in Section 4 and our analysis in RQ3 arrive at similar conclusions, which cross-validate our shared insights, despite differences in task formulation (captioning vs QA), attention computation (image vs instruction/evidence vs non-evidence patch), and model families.
>
> ## 3. Methodology
>
> **ViCrop** uses a **coarser, custom visual patch grid** (determined by $N$), and selects the optimal crop box using signals from **one specific layer of the model** ($m$), and **optionally one specific layer of the connector** ($k$). For each patch, this results in a **hard binary decision** (included or not included).
>
> **VEA** uses the model’s **native visual patching** and performs **soft highlighting**: we extract attention from **multiple layers** selected through profiling, **denoise** attention to mitigate irrelevant high-attention tokens (caused by attention sink \[4,5\], as suggested by other reviewers), and then **smooth** the resulting mask in pixel space to obtain soft highlighting.
>
> Additionally, there exist other works that leverage the MLLM’s internal attention from the initial inference step to guide visual enhancement (for example, baselines \[6,7\] used in our paper, from which we draw inspiration). We believe this general direction is promising, and **both your work and ours provide good solution from different perspectives.**
>
> **For the above reasons, we believe our works focus on different aspects and explore the problem of improving VLM/MLLM visual understanding from different angles**.
>
> ## An interesting shared observation
>
> We also have an interesting observation we would like to discuss with you. In your Fig. 3, BLIP-2 and InstructBLIP appear to exhibit a periodic layer-wise pattern. We observe a similar phenomenon in Gemma-3 4B/27B (our Fig. 9), whereas LLaVA and Qwen do not show this behavior. We have not found related discussions in the literature, and our current guess is that this may arise from differences in pretraining pipelines or visual encoding architectures.
>
> We would be very interested to hear whether you have any insights into what training choices or architectural designs might lead to this pattern. We understand this is not a central issue and the answer may lie in many subtle details, but exploring such phenomena from different perspectives is always enlightening.
>
> **Again, thank you for your interest and for leaving the public comment. This is our (the main author's) first ICLR submission experience, and it has been great to exchange ideas with a broader audience during the review stage. We hope our response addresses your concerns, and we welcome any further questions or thoughts..**
>
> Best,
> Authors

---

> > ### Author Response · Authors · 2025-11-20
> >
> > References
> >
> > > [1] Tong, Shengbang, et al. "Eyes wide shut? exploring the visual shortcomings of multimodal llms." Proceedings of the IEEE/CVF Conference on Computer Vision and Pattern Recognition. 2024.
> > > [2] Liu, Yexin, et al. "Unveiling the Ignorance of MLLMs: Seeing Clearly, Answering Incorrectly." Proceedings of the Computer Vision and Pattern Recognition Conference. 2025.
> > > [3] Chen, Cong, et al. "PerturboLLaVA: Reducing multimodal hallucinations with perturbative visual training." arXiv preprint arXiv:2503.06486 (2025).
> > > [4] Kang, Seil, et al. "See What You Are Told: Visual Attention Sink in Large Multimodal Models." The Thirteenth International Conference on Learning Representations.
> > > [5] Darcet, Timothée, et al. "Vision Transformers Need Registers." The Twelfth International Conference on Learning Representations.
> > > [6] Liu, Yexin, et al. "Unveiling the Ignorance of MLLMs: Seeing Clearly, Answering Incorrectly." Proceedings of the Computer Vision and Pattern Recognition Conference. 2025.
> > > [7] An, Wenbin, et al. "Mitigating object hallucinations in large vision-language models with assembly of global and local attention." Proceedings of the Computer Vision and Pattern Recognition Conference. 2025.

---

### Public Comment · ~Zaiquan_Yang1 · 2025-11-13
**Expect more clear clarification about some concepts.**

I appreciate the author's diligent work, which has given me some fresh insights. However, I have some confusion about the writing.

- I am confused about the introduction of RAPT, which deserves a more clear and formulaic description.
- In RQ2, how is the visual attention derived? Many works point out that the visual attention is less interpretative due to the vision sink tokens though the `LocalizationHeads`[A] has proposed a denoising solution to some extent. Besides, the approach seems similar to `Vicrop` which has been pointed out by original authors.


[A] Your Large Vision-Language Model Only Needs A Few Attention Heads For Visual Grounding, CVPR2025

---

> ### Author Response · Authors · 2025-11-20
>
> Hi Zaiquan,
>
> Thank you for your interest! Below are our responses to your questions.
>
> ## 1. Why introduce and use RAPT
>
> We use RAPT (relative attention per token) rather than total attention to avoid confounding effects caused by variable question lengths. Since different samples contain different numbers of text tokens, **total attention would make it unclear whether a model attends more to text simply because the question is longer**. Using attention per token allows us to compare the average importance of each image or text token fairly. The term “relative” refers to normalizing by the average attention over the entire input. This avoids issues caused by the extremely small raw attention values and provides a more interpretable scale.
>
> Formally, the image RAPT at layer $l$ is defined as $RAPT^{(l)}\_{\text{image}} := \frac{\sum \mathbf{a}^{(l)}\_{\mathcal{I}}/m}{\sum \mathbf{a}^{(l)}/n}$, where $\mathbf{a}^{(l)}\_{\mathcal I} \in \mathbb{R}^m$ is the attention vector over the $m$ image tokens, and $\mathbf{a}^{(l)} \in \mathbb{R}^n$ is the attention vector over all $n$ input tokens. Text RAPT is defined analogously by replacing the numerator with the attention over text tokens. Due to space constraints we did not include the full formula in the paper, as the exact definition does not affect the insights we aim to convey.
>
> ## 2. Deriving visual attention and addressing attention sink
>
> In RQ2, the derivation of visual attention follows the procedure described in Section 3.1. We indeed observe the vision/attention sink phenomena. **As discussed in Section 3.2 on attention mask denoising**, Figures 2 and 3 show that raw masks often contain scattered high valued patches unrelated to the true evidence. To mitigate this, we introduce a neighborhood filtering strategy that removes spurious activations and stabilizes the visual attention maps. **We will add clarifying discussion around vision sink near Figure 2 to improve readability. Thank you for pointing this out.**
>
> ## 3. Connections to ViCrop
>
> **For the distinction from ViCrop, please refer to our response to Jiarui. Specifically for the approach similarity you asked, please refer to section *3. Methodology* therein.**
>
> Again, we appreciate your thoughtful feedback. Please feel free to reach out with any further questions : )
>
> Best,
> Authors

---

### Author Response · Authors · 2025-11-30
**Rebuttal Summary**

Dear AC and SAC,

**We sincerely appreciate your time and effort in overseeing the review process for our submission. We understand that due to the special circumstances of this year, ACs are facing significant additional workloads, thus we would like to share a brief summary of the rebuttal to assist in your decision-making.**

# 1. We appreciate all reviewers’ recognition of our work’s strengths.

Three reviewers holds a positive attitude, even Reviewer 3b67 with rating 2 (has not yet participated in the rebuttal discussion) acknowledged that our work has good soundness and presentation. Our scores for soundness/presentation/contribution were 4/3/3/3, 3/3/3/3, and 3/2/3/2, respectively, before the rebuttal revision. In general, all reviews objectively acknowledged the core contributions of our work, including:
- (i) **The insightfulness of our findings**
    - RdjA - "valuable for guiding future research"
    - LvsH - "providing valuable insights for the community"
    - 8FW4 - "The phenomenon of “seeing but not believing” is intriguing"
    - 3b67 - "an insightful analysis of attention localization across layers".
- (ii) **The thoroughness of the analysis and experiments**
    - RdjA - "demonstrates effectiveness across multiple benchmarks"
    - LvsH - "a thorough investigation of how different layers in VLMs process inputs and distribute attention"
    - 8FW4 - "The experiments and ablation studies are comprehensive"
- (iii) **The simplicity, versatility, and effectiveness of the proposed method VEA**
    - RdjA - "can be applied to various VLMs with zero training cost"
    - LvsH - "simple yet effective, showing consistent improvements across eight different VLMs"
    - 8FW4 - "provides both interpretability and performance improvement"
    - 3b67 - "methodologically well-designed", "provides a simple yet interpretable tool for reducing hallucinations".

# 2. We conducted a detailed rebuttal which meaningfully improved the quality of our work.

Guided by constructive feedback, we revised serveral parts of the main text to improve readability, discuss related work, and elaborate on technical details. Furthermore, we added 6 pages of additional experiments, analysis, and discussion in the appendix to further enhance the comprehensiveness of the paper, including:

- **Broader Task Evaluation**: We extended experiments to multi-turn conversations and multi-image QA (Appendix C.1), as well as reasoning-centric benchmarks (Appendix C.2). Results show that our method consistently improves performance across these varied settings.
- **Deepened Analysis**: We tested the potential of multi-round self-validation with VEA and discussed possible future directions for dynamic reasoning (Appendix C.3). To understand when attention grounding might be inaccurate, we provided quantitative analyses on "when attention is not reliable" (Appendix C.4). We also included a discussion on seeming discrepancies with existing findings (Appendix C.5).
- **Robustness & Visualization**: Finally, we verified the benefits of model-specific profiling and the robustness of VEA to inaccurate layer choices (Appendix C.6). We also enriched the paper with more qualitative examples in diverse natural scenes (Appendix C.7).

# 3. We appreciate the engagement from reviewers and the community, and have actively responded to all comments.

We have prepared detailed responses addressing every concern (including questions and weaknesses) raised by each reviewer and have implemented corresponding improvements in the paper. **We understand that due to this year's circumstances, the reviewer responses are reverted, but we are confident that our comprehensive rebuttal addresses most, if not all, of their concerns.**

Additionally, we actively addressed public comments left by two researchers early in the review process. Jiarui suggested citing his prior work, ViCrop, where they focus on using attention to crop images for improving visual perception; we added a discussion in the related works section but respectfully disagreed with his statement that ViCrop is "highly similar," based on significant differences in motivation, analysis, and methodology between the two papers. Subsequently, Zaiquan asked about concepts and technical details regarding the analysis section, and we provided detailed explanations and clarifications to resolve his queries.

**We hope this summary is helpful in facilitating your decision. All points in this summary are based on objective information from the rebuttal. Once again, we thank you and all the reviewers for your thoughtful evaluation and for helping us improve this work.**

With best wishes,
The Authors

---

### Note · Authors · 2026-01-26

I have read and agree with the venue's withdrawal policy on behalf of myself and my co-authors.

---

> ### Note · Program_Chairs · 2026-01-26
>
> We approve the reversion of withdrawn submission.

---

### Meta-Review · Area_Chair_i26K · 2026-01-07

**Summary:**

The paper received three positive reviews (Ratings: 6, 6, 6) and one negative review (Rating: 2). The authors provided a very comprehensive rebuttal, adding significant experiments on multi-turn/multi-image settings and addressing the negative reviewer's concerns regarding attention interpretation. The negative reviewer (3b67) did not engage in the post-rebuttal discussion, but I find the authors' defense convincing. The public comment regarding a "highly similar" concurrent work was also adequately addressed; the methodologies differ significantly (soft attention modulation vs. hard cropping). Thus ,I tend to sugget a acceptance fgor this paper.

**Reviewer Concerns:**

--addressed--: As the author suggested, the answer from a single inference might not be accurate. The same applies to the attention map — can this method apply to the same question and image multiple times in a cascade way?

**Reviewer Scores:**

Regarding the comment on evaluating cases where the model does not attend to the correct regions, the authors provided additional experimental results during the rebuttal that explicitly address this issue. These results clarify how often the model produces correct answers despite incorrect attention and how performance is affected when highlighting incorrect regions. Based on these clarifications, the AC believes that the reviewer would likely have revised their score had they been able to participate fully in the discussion.

---

### Decision · Program_Chairs · 2026-01-26

Accept (Poster)